# The Role of Structure in Polymer Rheology: Review

**DOI:** 10.3390/polym14061262

**Published:** 2022-03-21

**Authors:** Valery G. Kulichikhin, Alexander Ya. Malkin

**Affiliations:** Institute of Petrochemical Synthesis, Russian Academy of Sciences, 29 Leninsky Prospect, 119991 Moscow, Russia; alex_malkin@mig.phys.msu.ru

**Keywords:** polymer, melts, solution, macromolecule, non-linearity, structure, instability, thixotropy, yielding, viscoplasticity, viscoelasticity

## Abstract

The review is devoted to the analysis of the current state of understanding relationships among the deformation-induced structure transformations, observed rheological properties, and the occurrence of non-linear effects for polymer liquids (melts, solutions, and composites). Three levels of non-linearity are the base for consideration. The first one concerns changes in the relaxation spectra of viscoelastic liquids, which are responsible for weak non-linear phenomena. The second one refers to the strong non-linearity corresponding to such changes in the structure of a medium that leads to the emergence of a new relaxation state of a matter. Finally, the third one describes the deformation-induced changes in the phase state and/or the occurring of bifurcations and instability in flow and reflects the thermodynamic non-linear behavior. From a structure point of view, a common cause of the non-linear effects is the orientation of macromolecules and changes in intermolecular interaction, while a dominant factor in describing fluid dynamics of polymer liquids is their elasticity. The modern understanding of thixotropic effects, yielding viscoplastic materials, deformation-induced phase transition, and the experimental observations, demonstrating direct correlations between the structure and rheology of polymer liquids, are the main objects for discussion. All these topics are reviewed and discussed mainly on the basis of the latest five-year publications.

## 1. Introduction

The huge variety of polymer-containing liquids used in modern technologies demonstrates great diversity of their rheological properties that corresponds to the requirements of their application areas. Therefore, it would be highly desirable to find out some fundamental principles determining these properties and quantitative rules for describing the observed effects.

There are two main concepts that can be the grounds for answering these challenges. The first one is the concept of the structure of the matter, and the second one is the idea of the non-linear behavior of a liquid. Although the term “structure” is not unequivocally determined, one can “feel” that it means the mutual arrangement in space of material elements of different dimensional levels and the various types of interactions between them. Regarding the role of non-linearity, the following quote accurately represents the idea: “It is difficult to make much headway in … rheology without an appreciation of the general importance of non-linearity” [1].

Meanwhile, it seems appropriate to classify non-linear phenomena into three main groups: (1) effects due to large deformations without changing the structure of matter (geometrical non-linearity); (2) changes in the structure with the emergence of a new order of a matter (physical non-linearity); and (3) induced instabilities of various types, leading to the transformations of the thermodynamic state of matter (thermodynamic or phase non-linearity) [2]. 

It should also be noted that structure phenomena are intrinsically related to time and require kinetic arguments for their description. Usually, the term thixotropy and/or rheopexy is used as an integral feature of time effects [3]. At the same time, it should be noted that time effects associated with linear viscoelasticity are also characteristic for polymer substances [4]. These effects are due to the dynamics of macromolecular chains and do not affect the macrostructure of polymeric liquids. Both time effects can be superimposed on each other and appear on the same very wide time scales, which can be characterized by the deformation rate or frequency. It is necessary to distinguish between the nature of the observed temporal effects.

Last, but not least, is the introductory comment: the structure in polymeric substances can exist in a static state created after sample preparation. Then, the structure can appear due to deformation (shear or extension) of the substance, as a result of deformation-induced self-organization or destruction phenomena. 

The current understanding of rheology is based on numerous approaches to linking these basic concepts to more general polymeric materials, as it remains reasonable. This review is an attempt to reflect the current situation in this field.

## 2. The Cox–Merz Rule as a Reflection of the Viscoelastic Origin on Non-Linear Rheology

A standard non-linear rheological effect in polymer systems is the non-Newtonian behavior presented by flow curves—the shear rate dependence of shear stress or apparent viscosity, $\eta \left(\dot{\gamma }\right)$. It looks rather amusing that non-linear flow curves look equivalent to the frequency dependence of the dynamic viscosity in a linear strain range. This is the so-named Cox–Merz rule [5].

This linear–non-linear correlation is expressed by the following equation:(1)$$\eta \left(\dot{\gamma }\right)=\eta *\left(\omega \right)$$
where $\eta \left(\dot{\gamma }\right)$ is the dependence of the apparent viscosity calculated by the Newtonian definition, which is applied to a non-linear domain of flow, and η*(ω) is the frequency dependence of the dynamic viscosity measured at low strains, where a matter demonstrates a linear viscoelastic behavior. The physical sense of this rule was discussed in [6]. One can find numerous examples of the applicability of this rule, including recent rather revealing data obtained for a series of polyethylenes with different molecular weights [7].

Two aspects of this rule are important to discuss. The first aspect hints at the origin of non-linearity as a consequence of viscoelasticity, and this is true for simple objects, which is a non-linearity of the first type. This approach was developed by demonstration of a direct correlation between the apparent viscosity in a non-Newtonian flow of a viscoelastic polymer melt and the evolution of its relaxation spectrum [8]. The following equation shows this dependence:(2)$$\eta \left(\dot{\gamma }\right)=\int_{0}^{\infty }\theta F\left(\theta ,\dot{\gamma }\right)d\theta$$
where the function $F\left(\theta ,\dot{\gamma }\right)$ presents the dependence of a linear relaxation spectrum F(θ) on the shear rate. The concept of the deformation-induced changes in relaxation spectrum was developed earlier [9] in a rather general form. 

The second aspect of this issue is presented by experimental data in Figure 1, which illustrates the relationship between the viscoelastic behavior of a polymer liquid (reflected by ratio of flow and recoverable deformations) and its non-Newtonian behavior [10].

The correlation between elasticity and non-Newtonian flow is shown in Figure 2, which shows the direct dependence of the apparent viscosity and the stored (elastic) energy in a wide range of the viscosity change.

The concept of linear–non-linear similarity in the rheology of polymer liquids also explains the well-known effect of spurt at high deformation rates. This is the viscous-to-rubbery (elastic) state transition, as shown in Figure 3. We meet here with a relaxation transition that is related to viscoelastic behavior at high deformation rates corresponding not to flow, but to the rubbery state of a melt. 

This transition starts from small-scale surface distortions of the jet (a) and passes through the large-scale periodic distortion (b) to elastic turbulence with fractures of the jet (c) due to high shear and normal stresses.

This transition is characterized by the dimensionless Weissenberg number $Wi=\dot{\gamma }\theta$ (where $\dot{\gamma }$ is shear rate and θ is a characteristic relaxation time of a viscoelastic liquid) and happens at Wi of the order of 1.

Figure 4 shows typical frequency dependences of the storage and loss components of the complex elastic modulus in the linear domain of polymer viscoelasticity. 

It is seen that in the transition to the rubbery plateau (G’ = const), G” decreases. Then in increasing the frequency, both components of the dynamic modulus increase. Thus, change in dynamic viscosity at increasing frequency is not monotonous, while apparent steady-state viscosity always decreases in increasing shear rate up to the onset of instability (like in Figure 3). It means the Cox–Merz rule becomes invalid at high frequencies (shear rates) that correspond to the transition to non-linearity due to the flow–rubbery relaxation transition.

A similar picture of the linear-to-non-linear transition takes place in extension. Figure 5 shows the deformation rate dependences of the flow and elastic strains expressed by the Henky measure ($\varepsilon^{H}$) [10,11,12].

One can see that the minimum of the total deformation $\varepsilon_{total}$ corresponds to Wi of the order of 1. At higher Wi values, the elastic deformation predominates, and flow becomes negligible. This is a typical flow-to-rubber transition equivalent to the transition detected on the frequency dependences of the complex elastic modulus. 

The above-presented example, as well as many experimental data in current publications, corresponds to geometrical or weak non-linearity. Nevertheless, even non-linearity of this kind can finally lead to rather strong effects, such as the elastic turbulence and melts fracture. Of course, a discussion of the applicability of the Cox–Merz rules at high strain rates (high Weissenberg numbers) becomes meaningless since in this case, we encounter flow instability. Figure 1, Figure 2, Figure 3, Figure 4 and Figure 5 correspond to viscoelastic non-linear behavior based on a concept of the elasticity of polymer liquids that was determined by the Weissenberg number. However, at Wi << 1, the weak non-linearity takes place. For at Wi >> 1, there is another type of non-linearity, which manifests itself as instability and a relaxation transition. 

The deviations from the Cox–Merz rule were described for complex fluids (e.g., multicomponent heterogeneous systems and interacting polymers) that exhibit deformation-induced structure transformations, even at low shear rates [13]. This is a case of deformations of polymer melts and solutions in extension if developing structural effects and inhomogeneous macroscopic deformations take place. Then, these phenomena lead to a change in the mechanism of flow, rupture of the extended specimen [14,15,16], and macroscopic displacements of the structure elements of a matter [16]. The recent publications have demonstrated more complicated behavior of polymer liquids in extension inconsistent with the traditional viscoelastic model [17,18,19,20,21]. As a result, a new model for understanding the origin of rupture in extension was proposed [22], which was discussed in [23,24,25]. This model is based on the concept of brittle fracture (at high deformation rates) starting from the rupture of primary C–C bonds in a polymer chain. It was supposed that the initial act of cracking occurs due to thermal fluctuations (entropic fracture), which leads to a short-time concentration of the strain energy as a cause of chain rupture. This effect is determined by finite extensibility of polymer chains.

All mentioned experimental results are inconsistent with the Cox–Merz correlation. The failure of the Cox–Merz rule reflects the limit of the weak non-linearity in describing and predicting rheological properties of complex matters, and it appears necessary to involve other mechanisms of these effects (e.g., thixotropy and the yielding behavior of polymer liquids), leading to the stronger non-linearity.

## 3. Thixotropy—The Most Evident Reflection of Structure Transformation

Structure aspects of the non-Newtonian behavior are the most evident in observation of the thixotropic behavior of complex liquids. Wo. Ostwald’s concept (1929) of “structural viscosity” *(**Strukturviskosität*) that was developed in the monograph *Thixotropy* by H. Freundlich (1935). This was one of the historical milestones in rheology. This phenomenon is understood as time-dependent changes in rheological properties not related to viscoelasticity or other processes occurring in time (e.g., polymerization). Meanwhile, it is not always easy to separate these two types of time effects. Formally, “pure” thixotropy is observed if a time-dependent viscous stress only takes place with stress relaxation or recovery (elastic) strain [26]. One can find a detailed discussion concerning the difference between thixotropy and other time-dependent processes in [27,28]. 

Moreover, at least in some cases, thixotropic rejuvenation happens only due to the viscous rate of deformation but not to the structural effects; this scenario does not violate the second law of thermodynamics [29]. Thixotropic behavior close to the “ideal” case is typical mainly for non-colloidal suspensions, such as paints, compositions used in the oil industry, construction and road industry, and numerous colloidal systems. However, polymeric liquids, especially multicomponent systems (solutions, blends, melts with solid filler), can also exhibit thixotropic effects.

The commonly used experimental approach to characterize thixotropy is measuring a flow curve of a substance and observing the viscosity bifurcations [30]. There are many experimental (mainly applied) publications, where a state of structure is considered the shear rate dependence of non-Newtonian viscosity $\eta \left(\dot{\gamma }\right)$, changing from the maximal value, $\eta_{\max}$, to the minimal value, $\eta_{\min}$:(3)$$\alpha \left(\dot{\gamma }\right)=\frac{\eta \left(\dot{\gamma }\right)-\eta_{\min}}{\eta_{\max}-\eta_{\min}}$$

This ratio is treated as a measure of the structure transformations in a liquid. Indeed, the shear rate is, in a sense, the reciprocal time. However, the non-Newtonian effect can have different causes, and Equation (3) cannot be considered a universal quantitative definition of the thixotropic effect.

The rather commonly used practice in evaluating the thixotropic behavior for industrial fluids is measuring viscosity in a scanning regime at increasing and decreasing shear rate. The effect of bifurcation is shown in Figure 6, and can be estimated by the area between the up and down curves treated as the work spent for changing the structure of a liquid.

This way of characterizing a thixotropic effect may have a technological application and even be standardized for special purposes but does not have a physically based ground. Indeed, the area between both branches depends on the rump of scanning as well as on the choice of the start and end points.

More expressive and representative are data obtained in measuring the viscosity evolution at different constant shear stresses (Figure 7). 

Shear-induced thixotropic transitions between different structural states are nicely illustrated by the bifurcation of the viscosity when self-oscillations occur in a certain range of given shear rates (Figure 8 [31]). Self-oscillations in shearing can be a consequence of different reasons, including the viscoelasticity of a substance, but in some cases, this effect is undoubtedly due to the rupture and coalescence of clusters in a liquid matrix, which was directly observed in experiments [32]. The formation of a cluster was even considered as a kind of phase transition [33].

The state of a thixotropic material is frequently characterized by some parameter *λ*, intuitively considered a measure of the “structure”. For example, the following simple kinetic equation was proposed for this parameter [34]:(4)$$\frac{d\lambda }{d\dot{\gamma }}=\frac{1}{T_{0}}-\alpha \lambda \dot{\gamma }$$

For thixotropic liquids, many different models have been proposed and discussed based on superposition of aging and shear rejuvenation that corresponds to structure formation at rest and breaking under shearing. The constant *T*_0_ is the rate of structure approaching its equilibrium level at “rest”, if $\dot{\gamma }=0$. The factor α specifies properties of the sample under shearing. Then, the apparent viscosity is assumed to be some unknown function of *λ*.

A more general kinetic model includes both side effects—breakdown and build-up of the “structure”—expressed by the parameter *λ* [35]
(5)$$\frac{d\lambda }{dt}=-k_{1}\lambda {\dot{\gamma }}^{a}+k_{2}\lambda {\dot{\gamma }}^{b}$$
where the first term reflects the rate of the structure breakdown (rate constant *k*_1_) and the second term the rate of the reverse process of the build-up (rate constant *k*_2_). Powers *a* and *b* are the orders of these processes, considered to be some kinds of “reactions” of the structure rupture and recovering 

More complete consideration of thixotropy relates this phenomenon with yielding [26] (see the next section). Then, today’s modeling thixotropy discusses this effect in combination with all other time-dependent effects and proposes models of elasto-visco-plastic behavior. The term thixotropic elasto-visco-plastic (*TEVP*) matters is widely used. Therefore, the rheological constitutive equation must cover both the structure regions of a viscoplastic fluid: below and above the yield stress. The apparent viscosity is considered a function of the structure parameter *λ*, and the generalized model also includes a Maxwell-type viscoelastic element [34]. The rheological equation is usually formulated for stress but not for the shear rate, because any structure has some strength, and it is necessary to apply stress to destroy it. 

This complex description of the state of TEVP fluids can be presented by a 3D map, and the conditions to the dominance different types of rheological behavior require the introduction of several dimensionless groups [36], especially taking into account that viscoelastic and thixotropic effects can occur in different time scales [37]. Moreover, it is quite reasonable to believe that different structures can coexist with their own kinetic characteristics of breakdown and build-up processes. In this case, instead of a single value of *λ*, the following assumption was advanced [38]:(6)$$\lambda =\sum_{1}^{N}C_{i}\lambda_{i}$$
where *C_i_* is a weight of the *λ_i_* structure.

Some other rather complicated expressions for the degree of structuring were also considered instead of (5) and then converted into a tensorial form [39]. This approach led to the construction of several dimensionless groups, characterizing the type of the rheological behavior of a matter [40] 

An increase in the number of adjusting parameters, surely, makes it possible to more accurately fit experimental data and describe the totality of various effects observed under different deformation modes. However, simultaneously, this makes it more and more difficult to look for experimental unambiguity in finding these parameters, which leads to serious limitations in applying any complicated model, including Equation (6), for the real practice. So, possibly, it is reasonable to stop at some intermediate position not chasing excessive precision but trying to find qualitative, or at least semi-quantitative, descriptions of experimental facts. 

The thixotropic effect, as well as the behavior of TEVP fluids, is usually modeled for simple shear, which gives a rather demonstrative presentation. However, it is not enough, e.g., for solving boundary dynamic problems, and for generalization for the 3D stress state, it is necessary to formulate the complete constitutive equations by including the kinetic factors such as those presented by Equation (5) [41,42,43]. 

Today’s understanding of the different manifestations of thixotropy (time-dependent yielding, hysteresis in shear-rate ramps, the role of rest time, non-monotonous flow curves, viscosity bifurcation, kinematic hardening, and so on) were considered and compared with different types of the proposed phenomenological constitutive models in the above-cited publication and analyzed in [43]. The authors of these publications rightfully indicated the limitations of the existing models and the necessity of examining transient and non-viscometric flows (e.g., extension). Meanwhile, the main problem is understanding what we exactly mean when saying “structure” and which microstructure transformations are responsible for different macroscopic phenomenon.

In this review, we do not list a great number of experimental studies because one can find many references in publications [26,27,28,29,30,31,34,36,44], thus, there is no sense in repeating them here. The flood of studies of thixotropic behavior in various technological liquids continues, due to the necessity to classify and standardize these objects. 

## 4. Yielding—Structure Breakdown Leading to Solid-to-Liquid Transition

The effect of yielding in viscoplastic fluids has already been mentioned in discussing thixotropic behavior. However, there are some additional features of strong non-linearity related to yielding. This kind of rheological behavior is crucially important for many areas of application, including composites based on polymer matrices.

In the Introduction to the collection of papers devoted to the centenary of advancing the concept of yielding in visco-plastic fluids (E. Bingham, 1916), it was stressed that it was a jump signifying a birthday of rheology [45]. Meanwhile, during these 100 years, a lot of new information was obtained, which highlighted new aspects related to yielding in structured substances and these allowed us to understand this phenomenon deeper and from new positions [46]. However, the main fundamental fact remained unshakable—yielding is directly connected with the deformation-induced structure transformation of complex fluids. 

In particular, yield stress along with surface tension enter the Bingham number for emulsions: (7)$$Bn=\frac{\sigma_{Y}}{\alpha /R}$$
where *α* is the surface tension and *R* is the radius of droplets in a liquid matrix. This criterion determines a possibility of the breakup of liquid particles in emulsions at stresses exceeding the yield stress [47].

Yielding in rheological complex fluids is a possibility for a material to exist in two mechanical states: a solid-like body at rest or at low stresses and a liquid state when exceeding some critical stress called the yield stress. In this sense, yield stress is a measure of the strength of a structure created in a material, and yield stress corresponds to the point of the stress-induced transition related to a jump-like change of the structure. 

Previously, the basic approach considered the yield stress as an independent constant, characterizing a complex fluid. Experimental measurement of the yield stress encounters some difficulties connected with a necessity to estimate a value at σ→0 that can be done only by extrapolation, accompanied with an inevitable lack of unambiguity [48]. Different experimental difficulties in finding $\sigma_{Y}$ were also discussed in [49,50]. Then, new experimental methods were proposed to obtain real values of the yield stress [51]. Different theoretical aspects and experimental problems occurring in measuring very low yield stresses were discussed in [52].

Nevertheless, the main reason of the difficulties in the $\sigma_{Y}$ measuring is time effects, which were briefly mentioned in the previous section devoted to the thixotropic effects happening in time. These effects show, in particular, that yield stress is not a constant inherent to a particular viscoplastic material because the point of the solid-to liquid transition depends on stress. This transition is described not by a single point, but it is characterized by the durability of the structure, i.e., by the dependence t*(σ), where *t** is the time corresponding to the solid-to-flow transition. An example illustrating the difference between the initial (Bingham’s) understanding of yielding and the concept of durability is shown in Figure 9. 

This transition can be also treated as the difference between rigid (a) and soft (b) structures. Deformations under stress occur in a solid state, though they are typically small for rigid structure, but can be rather high for soft structures. The latter is illustrated in Figure 10 for such media by a rather sharp break in the slope of γ(t) dependence [53,54].

Measuring in the range of low stresses and shear rates should determine whether this material is a viscoplastic fluid or this is a liquid and its flow curve includes the plateau of the maximal Newtonian viscosity. In everyday experimental practice, such measurements are performed in many cases, and a flow curve is a measure in the scanning mode of the changing shear rate. This is promoted by modern technique using the software that allows for using “comfortable” procedure without human continuous control. The comparison of experimental data obtained at different shear rates and at prolong measuring shear stress (or apparent viscosity) is shown in Figure 11 [55]. Let the length of a time step *t* at every shear rate be chosen as 10 s.

Comparing these two methods (continuous and scanning) of measuring shows the following [56]:

It is seen that the lower the shear rate, the longer the time it takes to reach a time-independent viscosity value. This might be treated as the limit of viscosity corresponding to the Newtonian plateau. However, in the scanning mode of measuring, all viscosity values at shear rates, lying on the right of the cross-point of the dotted line, predict the existence of the Newtonian plateau. Indeed, *t* = 10 s (at this example) is not enough to reach the steady flow that is a necessary condition for finding a correct value of viscosity. Then, the observed “viscosity” corresponds to the transient state and the apparent “viscosities” are the same, regardless of the shear rate as clearly seen in Figure 11. This creates a false impression that we have reached the plateau of the maximum Newtonian viscosity. The lower the shear rate or the longer the time of shearing, the higher the apparent viscosity seems, and finally it becomes infinitely high at the yield stress because “No steady state flows below the yield stress…” [57]. 

The latter means that the correct estimation of the viscosity can be reached only if the length of a step in scanning *t* is longer than ${\dot{\gamma }}^{-1}$ at least by the decimal order. Then the type of the rheological behavior at low shear rates can be established only at very long shearing. This requirement is not always met in everyday experimental practice. Discussing the experiments in the domain of yielding is very often combined with wall slip, which was observed in many studies; see, for example [58,59]. 

These difficulties in quantifying the yield stress raised the paradoxical question of “The yield stress myth” [60], and the equivalently paradoxical (and possibly, quite correct) answer, “Well, it depends on what you mean by … the yield stress” [61]. Now, we do not doubt the existence of the region of the solid-like behavior in viscoplastic fluids. Indeed, the start-up deformation of viscoplastic media are elastic [62,63]. The direct experimental evidence is the frequency-independent storage modulus (Figure 12), like for any solid.

This is completely true for rigid structures, while analogous dependence for soft structures can have a slight non-zero slope that is a reflection of some relaxation processes in these media.

Taking into account the experimental difficulties of determining the yield stress for practical applications, it is necessary to standardize the experimental procedure for measuring some quantity, which is interpreted as the yield stress, although, surely, it is not enough for fundamental research, requiring more rigorous and different approaches.

Most of the applied and theoretical studies of viscoplastic fluids are carried out in shearing. The non-linearity happened due to structure breakup at shear stress corresponding to a solid-to-liquid transition. Meanwhile, non-linear effects always lead to the 3D stress state that is well known for viscoelastic liquids. Generalization of the shear non-linearity for 3D deformation is important for constructing a rheological constitutive equation and its application to non-viscometric flows (e.g., extension). The problem of the correct construction of a non-linear 3D equation was discussed in detail in some fundamental monographs (see, e.g., ref. [13]).

A general approach to yielding should be based on the formulation of a constitutive (tensorial) rheological model, such as that proposed in [64]. The convincing experiment demonstrating non-linearity in shear is the normal stresses appearance usually connected with the elasticity of liquids (the Weissenberg effect). Direct measurements have showed that at low shear stresses (below the yield stress), normal stresses are constant and do not depend on the shear rate [65]. These normal stresses are not related to elasticity but are likely explained by the Reynolds dilatancy. As discussed above, the current understanding relates thixotropy with yielding in a model of elasto-visco-plastic fluid. Then normal stresses should be a general feature in the flow of such fluids at stresses exceeding the yield stress as well as due to elastic deformation at low stresses, below the yield stress. This is clearly shown using different models of viscoplastic fluids in [66]. 

Experimental studies based on this approach are very rare. The work carried out on non-thixotropic liquids [67] was the first systematic measurements of normal stresses in shearing. This study showed that both normal stress differences are proportional to the square of shear stress in the full range of them. This is similar to the properties of many viscoelastic liquids [56]. 

The non-linearity of yielding liquids is also well detected by large amplitude oscillations because such a method of testing touched a transition through the yield stress and this led to bifurcations [68]. The yielding of viscoplastic fluids can be detected not only in shearing, but in other types of deformation. Yield stress for 3D stress states should depend on the geometry of deformation. Indeed, yielding was observed for extension [69], and it convinces that the effect of yielding is the 3D phenomenon, and the value of the yield stress should be presented as a tensor [70,71]. The theoretical consideration assumed that the critical point for the solid-to-liquid transition in the 3D stress state is determined by the von Mises criterion quite like the rupture of solid material. This criterion is written as
(8)$$\sigma_{Y}=\sqrt{\frac{1}{2}\sum_{1}^{3}\sigma_{n}^{2}}$$
where $\sigma_{Y}$ is the shear yield stress measured and $\sigma_{n}$ are three principle stresses. This equation shows that the critical point of the solid-to-flow transition is determined by the second invariant of the stress tensor, which is calculated including normal and shear stresses.

Then the yield stress for extension $\sigma_{Y}^{n}$ can be estimated as
(9)$$\sigma_{Y}^{n}=\sqrt{3}\sigma_{Y}$$

However, experimental studies showed that the observed $\sigma_{Y}^{n}$ data are higher than predicted by Equation (9). This was explained by non-homogeneous deformation in extension, leading to the difference between local rheological properties and the bulk behavior of an object, characterized by $\sigma_{Y}$ [72].

## 5. Solid Particles in Polymeric Liquids—Basic Model of Multicomponent Media

### 5.1. A Single Particle in a Liquid Matrix

While discussing the structure–rheology relationship, we must realize that this structure is created by deformation because measuring the rheological properties requires applying external stresses.

There are many structural effects in the flow of polymeric liquids observed experimentally. These effects are the most expressive if we examine structures of multicomponent mixtures. Partially, such systems are dispersions, and it is rather instructive to follow the deformation of an individual dispersed droplet/particle, including as a limiting case a macromolecular coil. 

The orientation and formation of an anisotropic microstructure under deformation were demonstrated by small-angle light scattering method for emulsions [73]. A complete theory of orientation and shear-induced anisotropy was developed in [74]. Solid (non-deformable) particles like those shown in Figure 13 rotate and also orient in the shear flow. 

The rotation and deformation of anisotropic particles lead to some non-linear effects, such as non-Newtonian behavior and the emerging of normal stresses. These effects are related to the first type of weak non-linear phenomena. The shape of anisometric particles affects the viscosity of the dispersions. The dependence of viscosity of spherical particles on concentration is described by the famous Einstein law with constant intrinsic viscosity [η], equal to 2.5. However, it is often forgotten that [η] for anisometric particles is not constant but depends on the degree of anisotropy. Figure 14 presents the results of rather old theoretical calculations of the dependence of [η] on the D/d ratio of the particle. This is the simplest demonstration of the structure role in the rheological properties of dispersions regardless of its nature.

The anisodiametric particle is a popular model of macromolecule. Indeed, polymeric molecules orient in different geometries of flow—stronger in extension than in shear, but the induced anisotropy is a necessary consequence of the flow. This structure rearrangement leads not only to creating anisotropic properties, but influences the shear viscosity as can be understood from Figure 14. Additional rheological effects arise if dispersed particles are soft and can be deformed in the flow. According to this model and experimental data obtained for polymeric particles, this structure peculiarity is directly linked with shear thinning of a medium [75,76].

### 5.2. Self-Assembling in Filled Polymeric Liquids

When considering a multicomponent system with many randomly scattered components, the self-assembling is one of the most frequent and interesting phenomena. General aspects of this issue were discussed in [77,78]. This is a case of cooperative behavior demonstrating a chaos-to-order transition as an example of strong non-linearity in the rheology of multicomponent systems. Effects of such types are known for viscous and viscoelastic matrices and for the disperse phases of various nature. Below, some typical examples of the self-assembling of solid particles are presented.

There are many earlier observations demonstrating that the initially randomly distributed particles are able to implement self-organization, consisting in the formation of regular chains or strings (see, for example, [79,80]). This phenomenon is shown in Figure 15, where the arrow shows the transition from random to linear aligning due to shearing. The effect can be so strong that the newly formed structure at long enough shearing can look like a 2D crystal [81].

Rather expressive self-assembling was demonstrated by video monitoring of the particles movement in the rotational device, which finally resulted in the formation of a system of regular circles of disperse particles in a polymer matrix at a constant shear rate (Figure 16 [82]).

There is a certain link between the rheological properties of a polymer matrix and the self-organization effect. The elasticity of a polymeric liquid expressed via normal stresses in shear flow is initially considered a main reason of the particles’ alignment [83]. Further detailed experimental analysis showed that the viscosity thinning does not play the dominant role in the effect under discussion, but the elasticity can promote the string formation [84]. 

Dependences of shear stress, normal stress and viscosity on shear rate for the dispersion of Na-montmorillonite in PIB solution presented in Figure 17 allow us to combine the rheological response with a stream morphology in the operating unit of the sphere-plate type. The formation of the closed circles either in the neat melt (bottom photo) or in filled dispersion (middle photo) occurs under the condition of a prevailing normal stress (zone III). At constant stresses (zone IV), spurt behavior takes place, caused by the transition of the melt to the rubber-like state and its destruction (top photo). This example demonstrates clearly a role of elasticity in the ordered rings formation. 

Meanwhile, orientation remains an obligatory factor influencing the structure and consequently the rheological properties of a liquid matrix with dispersed solid particles. The evolution of solid particle orientation follows the development of deformation. Using the light scattering, the butterfly-type light scattering pattern for ordered morphology attributed to the formation of aligned structure was observed [85]. 

In the current literature, there are examples of direct correlation between the structuring and rheology of polymer substances, i.e., the experiments demonstrated that microstructural origin was an immediate cause of not only the shear thinning [86], but even the absolute values of viscosity. The most striking example of such behavior is the transition of solutions of rigid-chain polymers to the liquid crystalline (LC) state. The effect of reducing the viscosity with increasing solution concentration was observed in many cases (see, for example, ref. [87]). This phenomenon initiated a lot of not only scientific publications, but also patents [88], which gave start to the production of the new type of super-strong chemical fibers entitled Kevlar, Terlon, Armos, Rusar, etc., which are rather popular as the reinforcing phase in composites. 

The main reason for the decrease in viscosity is the transition of rigid-chain macromolecules to the LC state, i.e., one- or two-dimensional ordering causing anisotropy of rheological properties: low viscosity in flow direction and high in transversal one [89]. Without touching on some details of rheology of LC solutions and melts, let us briefly consider the behavior of a very simple example of the LC system—aqueous solutions of hydroxypropylcellulose (HPC) filled with anisometric clay particles, also capable of forming mesophases ([90]). This example is interesting due to the combined use of the rheological method and X-ray scattering. For such a complex system, a combination of these two methods is especially fruitful for correct understanding the real, but not hypothetical, structural changes. First of all, it is reasonable to show concentration dependence of the viscosity for neat and filled solutions (Figure 18). 

As expected, the presence of filler led to an increase in viscosity, but the behavior of concentration dependence remained the same as for a neat solution, i.e., with a maximum at the formation of the LC phase. Using a Couette operating unit with X-ray beam transversal to the shear direction, it was possible to distinguish the orientation and ordering of both crystalline components: LC matrix (LC solution of HPC in water) and layered alumosilicate filler. It is the most interesting to trace the evolution of diffractogram over the time of shearing. These data are shown in Figure 19 for the shear rate of 4.7 × 10^2^ s^−1^. Up to 70 min of shearing, the HPC reflex is located on the meridian and the clay reflex on the equator. Longer shearing is accompanied by the movement of the clay reflex from the equator to the meridian, which means transformation of its structure from a columnar to discotic one. Thus, the mechanical field induces such a transformation, shown in the rheological properties [90].

The final result is shown schematically in Figure 20. 

In addition, the shearing of suspensions can cause specific self-assembling of the filler particles, which can lead to the formation of a layered structure, as shown in Figure 21. The rheological consequence of the latter is the flow of a liquid between neighboring layers. These experimental results explain the meaning of the “minimal Newtonian viscosity” $\eta_{\min}$, which is frequently considered the limit of the complete breakup of the structure (see Equation (3)). Actually, this value is the viscosity of a liquid matrix (layers), which can be very low (depending on the nature of a matrix), but it is not the viscosity of any structured suspension in general.

The formation of layered oriented structures in polymer suspensions subjected to simple shear was also observed and proven by numerical simulation [91]. Modeling demonstrates a possibility of discontinuities and inter-layer displacement of sufficiently large blocks of solid particles as was earlier observed in [92]. This means that a system becomes macro-heterogeneous as a result of deformation (Figure 22). The consequence of heterogeneity is the undetermined motions of disperse particles at a high concentration of the disperse phase-sliding clusters relative each other [93,94]. A similar effect was observed for concentrated emulsions.

The effect of layering in multicomponent systems and the flow a matrix liquid between the layers is close, by nature, to shear banding, which is related to the separation of components. This is a strong non-linear effect. For measuring the rheological properties of such an object, it is important to take into account the heterogeneity of a sample and not treat the macroparameters obtained in standard rheometers as some “average” values for a sample under study.

Suspensions of hard particles can demonstrate viscoelastic behavior, though the Cox–Merz rule does not work in this case [95]. This means that the non-linear effects in the flow of such systems are due to structure rearrangements. Indeed, the direct observations made in the oscillatory regime of deformation of suspensions have demonstrated microstructure reorganizations. The visualization demonstrated that shearing destroys the particle clusters and the shear-induced structure modification decreased particle collisions and energy losses that led to Cox–Merz correlation failure. An increase in the amplitude of deformation strongly enhanced this effect [96]. These experiments have clearly proven the role of thixotropic structural transformations in the non-linear behavior of liquids. By modeling this process, it was also shown that a change in the interparticle interaction observed in rheological measurements results in very strong non-linear effect, such as the irreversibility of an initial structure and developing the structural chaos [44]. 

For highly filled compositions with different matrices (including polymer melts), the final critical stage of the structure formation leads to the phenomenon called jamming. The destruction and agglomeration of particles in filled polymer is a typical thixotropic process with its own kinetics [97], although jamming can be irreversible. There are many experimental observations of a sudden or continuous increase in the viscosity with shear rate for compositions containing high concentration of solid particles, ended by jamming when steady flow becomes impossible (Figure 23). Earlier papers [98,99,100] (among many others) contain typical examples of this phenomenon, while the review [101] summarized later publication in this field.

It is quite evident that thickening and subsequent jamming lead to a closer distance of the structure-forming elements and an increase in the interparticle interactions including friction that also gives its input [102]. Very often, the formation of contacting clusters can be seen in the microscope.

At a critical concentration, the solid structure elements create such a dense packing that there is not enough free space for their reversible flow, and the medium becomes elastic. Meanwhile, the packing of structure elements remains chaotic and there is enough free space for plastic deformations. So, we meet with elastoplastic (without flow) behavior of a subject as shown in Figure 24 [103,104]. This structure rearrangement also has a thixotropic character.

It should be emphasized that this is a different type of irreversible deformation than flow since the amount of plastic shear depends on the applied stress but not on the duration of its action. Compositions of this type (thermoplastic media with high content of solid component) are also thixotropic, but their structure after shearing becomes kinetically frozen.

## 6. Shear-Induced Structure Formation in Polymer Melts and Blends

### 6.1. Shear-Induced Self-Assembling in Polymer Melts

Deformation promotes the formation of precursors of regular structures [105]. 

The small angle neutron scattering for the tri-block copolymer (styrene-butadiene-styrene) demonstrated that shearing promotes the formation of a regular structure of micelle-like polystyrene blocks. This structuring was associated with the rheological behavior of the sample [106]. 

There have been many earlier publications demonstrating this phenomenon. Interest in this subject continues to exist, given the direct relationship between rheological behavior and creating morphology [107]. Numerous effects were found in this field, such as a role of branching in the morphological transformations [108], the role of crystallization nuclei in the formation of various structures [109,110], the effect of molecular weight [111], the quantitative characteristics of the shear-induced crystallization kinetics of certain polymers [112], and the role of the progressive association of the hard-segments in shear-induced crystallization of copolymers [113]. 

The shear-induced crystallization happens not only at stationary shearing, but also under large amplitude periodic deformations [114]. The shear-dependent kinetics of crystallization has a central line in these studies. Increase in the shear rate usually accelerates the rate of crystallization; sometimes, the reverse effect of weak slowdown in crystallization was observed and explained by the addition of self-nucleation agents, which promoted crystallization of less-entangled polymer melts [115].

Due to shear, crystallization happens simultaneously with the orientation of macromolecules [116]. The orientation limits the conformational freedom, makes macromolecules closed, and promotes the creation of more stable assemblies. The latter act as nuclei of crystallization. This picture was confirmed by the molecular dynamics method [117]. The non-equilibrium molecular dynamics simulation has also shown that there is a correlation between the crystal nucleation rate in paraffin melts (both in shear and in extension) and invariants of the extra stress tensor [118]. The temperature which determines a possibility of the formation, either of such types of structures or the super-crystal morphology (spherulites), is related to homogeneous (sporadic) or heterogeneous nucleation [119]. 

### 6.2. Layered Flow in Polymer Mixtures

Multicomponent polymer liquids, such as colloid systems, can also form shear bands, even if the difference between the components in concentration or component content is rather slight. Even a mixture of two fractions of the same polymer can separate into bands, and the dominant flow occurs in the melt of the low-molecular weight fraction [120,121], although the observed velocity profiles can be related to transient states of flow [122]. Nevertheless, the rheological measurements clearly reflect the structure transformation in the flow of polymer blends. The effect of such a kind has been a subject of intensive study during the last 20 years. The current situation in polymeric systems exhibiting strong non-linearity accompanied, for example, by non-local effects was described in [122]. This phenomenon was analyzed in detail for shear banding as a manifestation of non-local inhomogeneity [123]. 

The character of the flow of multicomponent system consisting of layers of different polymers was studied in earlier publications during the coextrusion of alternating multilayer films [124,125]. The authors observed the break in the velocity profiles at the interface (Figure 25).

Slip began at the shear stress exceeding some threshold, while the interface adhesion was significantly lower than one measured in the equilibrium conditions. This was explained by the reduced entanglements at the interface, that is also considered the main reason of the polymer–solid surface slip [126]. A decrease in the shear viscosity in multilayer constructions is due to interfacial slip, though the bulk viscosity grows in increasing the number of layers in the composition [127]. A decrease in the apparent viscosity is of a special interest in relation to the newly developed technology of coextrusion in fabricating the forced assembly of multilayer films [128]. 

The structure–rheology correlation is different for the extrusion of well-mixed polymers. In this case, there is no macro-phase separation, but the formation of numerous extended fiber-like parts of one polymer in the matrix of the other polymer takes place (Figure 26) [129].

Analytical SEM pictures combined with spectroscopy on Auger electrons give a rather expressive structure organization in the extrusion of a two-component blend of polysulfone (PSF; sulfur atoms are marked by green), and LC polyester (LCP; oxygen atoms are marked by red) (Figure 27) [130]). One can see a quite random distribution of the components after mixing (left) and their layered distribution as a result of the shear flow (right). The low-viscous LC component forms longitudinally elongated fragments in the isotropic matrix. In addition, LCP forms a thin surface layer that can be a result of being pressed out of the low viscous LCP on the conical section at the entrance to a capillary. Interestingly, the viscosity of a blend is rather close to the viscosity of the LC phase, which is much lower than the viscosity of the thermoplastic polymer [130].

The observed situation is very close to the earlier studied so-called composites in situ [131,132]; the authors explored an idea to create a composite from two liquid phases: if one of them is capable to form thin jets, they are converted into strong fibers after cooling. The above presented experimental data based on a new method of analytical SEM indicate that the initial state of mixing in the cross section of the filament does not change (at least, significantly) after deformation. Only the orientation of the components changes along the direction of flow. 

Meanwhile, there are theoretical arguments based on the two-fluid model, which predict the possibility of demixing, i.e., shear-induced phase separation [133]. Indeed, an effect of demixing was observed in the flow of polymer solutions (see below). In some rather old publications, the changes in molecular weight distribution through the cross-section of a capillary with the migration of low-molecular weight components to a capillary wall took place [134]. However, these observations were not met in later publications, and the random distribution of the component in the mixture remained unchangeable [130]. So, till now, convincing experiments showing the demixing in polymer blend are still unknown. The situation is somewhat different in polymer melts filled with solid fibers. In this case, the orientation of the fibers leads to the creation of an anisotropic structure, and its viscosity also becomes anisotropic [135]. The same can be expected in polymer blends, if at least one component orients at flow. 

Rheological characteristics of polymer blends creating space structures should depend on this structure, but so far, the rheological model for calculating their flow dynamics is still absent. Meanwhile, the understanding and quantitative description of the structure transformation in the shear flow is important for modeling polymer processing. 

## 7. Deformation-Induced Structure Effects in Solutions

### 7.1. Main Experimental Observations

The shear-induced structure–rheological effects were observed and described in many earlier studies of solutions of flexible-chain polymers. Thixotropic and other time-dependent effects were usually explained by the directly observed aggregation of macromolecules existing even in a very dilute solution [136]. The strong thixotropic effects are observed at the temperature up-and-down scanning in solutions of semi-stiff polyamidebenzimidazole (PABI) in DMAc, as illustrated in Figure 28 [137].

A solvent also enters these temporal structures. In many cases, the solvent–polymer interaction is crucial to the rheological behavior of the solution. For example, poly(acrylonitrile) is well soluble in dimethylsulfoxide, forming the viscous solution. However, adding a small portion of water to the solvent leads to rapid gelation. Surely, the rheological behavior in these two cases corresponds to quite different types of matters—solution or non-flowing soft system. There are many such catastrophic changes of the rheological behavior obliged to the specific polymer–solvent interactions.

These phenomena are of great importance for various branches of industry, including food production, pharmacy, oil technology, fiber spinning and so on. Such effects are quite similar to those observed for colloid systems and polymer melts, as were discussed above. Today’s main interest in the choice of a solvent and time effects in polymer solutions is related mainly to the technological application (see, for example, [138]). Therefore, the standardization of methods for assessing the corresponding effects is valuable in the first place.

The orientation of macromolecules under deformations seems rather obvious. Nevertheless, it is spectacular to confirm this by a direct experiment. Figure 29 shows the effect of birefringence in shearing accompanied by a flow curve, which is analogous to that shown before for the aromatic PABI solution. The main reason is the photoelasticity of the initially isotropic solution under action of mechanical stresses, but in this case, the change of color depends on the shear stress. Traditionally, the brightness varies only, but not the color. Since aramide macromolecules are rather rigid, it is possible to expect the molecular orientation reached at shear. In other words, such behavior is reasonable to interpret as the forerunner of LC ordering. This explanation seems to be real, because at higher concentrations in the other solvent, solutions of this aramide form the lyotropic LC phase.

Very similar research was fulfilled earlier on polyterephthalamide of p-aminobenzhydrazide X-500, which relates to semi-rigid polymers and are not capable of forming the LC phase in solutions. Nevertheless, the fibers spun from its solutions in DMSO have high mechanical properties. Based on the rheological data, the authors suggested that the transition of concentrated solution to the LC state can be carried out under the action of mechanical stresses realized in the spinneret holes. This approach considers mechanical action as an additional thermodynamic parameter that allows such a transition to occur during fiber spinning [139]. 

However, there is the other area of interest in the deformation-induced structure transformation in polymer solutions. This is the extension of solution jets for fiber spinning, and we should not touch on shear but on extension-induced transitions. A parallel study of shear and extension regimes, allowing them to be compared, can be also useful. Extension is the better method for creating the molecular orientation that promotes the assembling of parallel chains. The situation with the structure–rheology correlations is the most interesting for rigid- or semi-rigid chain polymers due to the parallel stacking of rod-like molecules resulting in the formation of the LC structure that is equivalent to the phase transition from chaos to 1D or 2D order. 

The universal master curve describing the dependence of the limiting strain (corresponding to the break of a sample) on the strain rate (Figure 5) relates to the linear behavior of polymers [11]. This master curve allows for predicting the deformation-induced transitions between different physical (relaxation) states of a polymer. From experimental point of view, the linearity means that a material should be unchangeable in the whole range of strain range, i.e., its structure should not feel applied stresses. Then, the time dependence of compliance should not be dependent on stress. Meanwhile, experiments show a more complicated picture. Two real types of behavior of polymer melts and solution in extension are shown in Figure 30. A solid smooth line with an exit to a plateau corresponds to the linear mode of behavior. Figure 30a demonstrates the effect of stain hardening and the second plateau at high strains (Figure 30b). Various types of the elongation behavior were discussed in [140]. 

It should be noted that the deviation from the linear behavior begins the earlier, the higher the specified strain rate. The extensional viscosity in the linear limit corresponds to the Trouton law: (10)$${\left(\sigma_{E}/\dot{\varepsilon }\right)}_{\dot{\varepsilon }\to 0}=3\eta_{0}$$
where $\sigma_{E}$ is normal stress, $\dot{\varepsilon }$ is strain rate, and $\eta_{0}$ is the maximal Newtonian viscosity.

The non-linearity in this case can appear due to the orientation dependent on the rate of deformation. Large deformations in the non-linear regime can lead to two structure effects. The first one is irreversible structure changes [14] and necking as a consequence of the continuous transition from a wide to narrow cross section (Figure 31a). The second case consists in the gradual decrease in the diameter of a sample (Figure 31b) happening due to the flow of a liquid. However, inhomogeneity of the cross-section takes place along the length of a sample in both cases. Therefore, the processing of experimental data is difficult since it is not obvious to which cross section of the specimen the measured force is assigned. Meanwhile, it is possible to find the rheological properties following the specified position on a liquid filament [141].

The necking formation in stretching viscoelastic polymeric liquids is a typical non-linear effect related to the deformational instability [142], and this is a particular case of the elastic turbulence [143]. The formation of periodic morphological fragments in a filament at stretching is a rather special and very interesting kind of non-linear instability. This phenomenon is called beads-on-string and an example is shown in Figure 32.

Many authors have observed structures of this type, including the occurrence of a hierarchical structure with the smaller drops between larger ones; the latter can be the result of a sequence of instabilities [144,145,146,147]. Initial understanding this phenomenon treated it as periodic relaxation of macromolecules. The thin parts of a filament connecting beads consist of fully extended chains, and the solvent collects inside beads, where the macromolecules relax and return to a coiled conformation. Indeed, the difference in properties between the solid-like parts of a filament (where at the final stage of stretching the macromolecules are supposed to be fully extended) and drops was proven [148].

Meanwhile, the other mechanism is advanced and widely discussed. It was noted that the initial bead-on-string structure at the final stages of stretching transforms into the separated droplets of a solvent on the surface of a filament [149]. This effect is called blistering, and this is a consequence of the space phase separation.

The phenomenon of deformation-induced phase separation is one of the most intriguing effects in the structure–rheology relationships. This effect is especially expressive for polymer solutions and closely related to the formation of periodic structures in extension. The phase separation starts with the appearance of random giant concentration fluctuations increasing at stretching that leads to wringing out a low-viscous solvent onto the filament surface. Figure 33 shows the initial stage of the occurrence of a periodic structure in stretching polymer solution, which consists in creating a thin solvent layer on the surface of an oriented polymer filament [150]. Then the solvent moves along the filament due to the capillary forces, and this flow creates the geometrical structure called unduloids, which pull together to form separate drops.

The final stage of this process is shown in Figure 34, where is seen the start of phase separation manifested as a color boundary due to intensive light scattering on the concentration inhomogeneities. Individual drops are definitely pure solvent, which can be easily detached from the oriented filament [151].

This effect can be used in a special process in the fiber technology called “the mechanotropic spinning” [152]. 

### 7.2. Theoretical Argumentations

There are two commonly accepted approaches to understanding experimental data obtained in non-linear modes of extension. First, it is possible to use different macroscopic rheological constitutive equations. One can find a typical example of the theoretical analysis of the necking instability in stretching [153]. Second, discussion is based on micro-level considering intermolecular interaction as a result of deformation of macromolecular coils due to orientation [154]. 

In many earlier publications, such parameters as viscosity and surface tension have been traditionally examined. Now, the role of viscoelasticity is included in theoretical and experimental examination [155,156]. Just the elasticity should be considered the driving mechanism for the emergence of deformation-induced regular structures including such as a bead-on-string formation [157]. The experiments showed that by varying rheological properties of polymer solutions, it is possible to realize transition from capillary to viscoelastic instability [158] and from the elasto-capillary mechanism of instability to phase separation [159]. At present, the general belief is that self-assembling in stretching polymer solutions is determined by the elasticity. Interestingly, it was found that the shape of the interface between a cylindrical part of a filament and a drop is quite similar to those of viscoelastic fluids and soft elastic solids, and this similarity stresses the decisive role of elasticity, even for liquids [159].

However, it is quite possible that the two outwardly similar beads-on-string and blistering have different origins of non-linearity. Indeed, they are different, as demonstrated above. The first one is a classical non-linear dynamic instability leading to an emerging regular surface structure, while the second one is phase separation related to the stress–concentration coupling, and mechanical force serves as a thermodynamic parameter [160].

A lot of experimental observations have shown that both shearing and elongational flow of polymer solutions can lead to phase separation at temperatures that differ significantly from the equilibrium points on the phase diagrams.

This effect can be hardly treated on the base of standard thermodynamic arguments. Perhaps phase separation can be understood, assuming that a solvent squeezed out a solution that likely takes place at a decrease in the cross section of a tube, thinning a filament, or the flow in curvilinear of channels. However, this approach was not cast in a rigorous theory. 

The concept of stress–concentration coupling based on a two-fluid model is more popular and widely accepted for explanation of the effect of the deformation-induced phase separation in shear and extension. The starting assumptions of this approach consider a two-component material (either a blend of two polymers or a polymer solution) as a mixture of two liquids. The concentration fluctuations arise in such a mixture, and the rate of their diffusion depends on stresses, which promote an increase in the concentration difference and thus lead to the phase separation. This model was described in detail for extension of polymer solutions [161], and it was also incorporated into a Rolie–Poly constitutive equation that allowed for predicting singularites (shear banding) in flow [162]. However, the band separation is an effect of flow, and therefore, something different from the phase separation discussed in the two-fluid diffusion model. Meanwhile, the following paradoxical situation exists: “instability is not thermodynamic in nature, for all practical purposes it seems to behave as though it were” [163]. This paradox does not yet have a general decision, though the cited work showed that an asymptotic limit of the two-fluid model could be linked with the Lyapunov functional, which is considered as a non-equilibrium analog to the free energy.

A very popular tube model (initially proposed by Doi and Edwards, 1986) and its modifications are widely used for analysis of the rheological behavior and instability of polymeric liquids. Among various versions of this approach, the above-mentioned Rolie–Poly model (developed in [164]) works quite satisfactorily in many cases, correctly describing all observed normal stress vs. the strain dependences in a filament extension [165]. The modern state of this type of modeling was considered in a comprehensive review [166]. Using an appropriated rheological model and general balance equations, it is possible to solve various dynamic situations in the flow of polymer solutions, for example, to consider the mechanics of jet formation [167,168]. For example, the boundary conditions favorable for the formation of the beads-on-string structure were discussed and established for viscoelastic liquids with the rheology described by the Oldroyd-B and Giesekus models [169]. 

Thus, there are some macroscopic models (first of all, reptation, then Rolie–Poly, Oldroid-B, Giesekus, and FENE) which are used to simulate non-linear effects and the occurrence of singularities in the flow of viscoelastic polymeric liquids. It could be argued whether these models are more physical or phenomenological ones. Nevertheless, the most advanced of them allow for correct describing the rheological behavior of viscoelastic polymeric liquids in the extension and to link their behavior with the architecture of macromolecules (but not with supramolecular structures).

Consideration of the physics of intermolecular interaction, which changes due to polymer uncoiling and orientation, is the second approach that allows us to understand the nature of non-linear phenomena observed when the polymer solution is stretched. The underlying concept concerning the peculiarities of elongation should take into account the real local concentration of macromolecules in extension [170]. Indeed, macromolecules in dilute solutions do not come into contact with each other and cannot create space structures in shearing. However, a solution with the same very low concentration can form stable (on some time scale) fibers in extension that means a realization of intermolecular contacts [171].

A coil–uncoil transition in extension results in increasing contacts between neighboring macromolecules and intermolecular interactions. This leads to changes in the rheological properties of a solution. So, we meet with a direct molecular structure–rheology correlation [172]. The effect of stretching on macromolecular interaction was characterized in terms of the coefficient of molecular friction depending on concentration and a mode of deformation [173]. Theoretical estimations based on Brownian dynamics simulation have shown that the stretching-induced uncoiling of macromolecules results in a hydrodynamic scenario depending on the concentration of flexible-chain polymer solutions [174]. 

Examination of the evolution of deformation-induced intermolecular interactions allows for describing the effect of phase separation in strong mechanical fields [175,176,177]. This approach is based on the thermodynamic principles, which take into account the orientation-dependent excluded-volume interactions, whereas attractive interactions are assumed to be conformation independent. However, the consideration was limited to the case of extension of dilute polymer solutions, where there are no entanglements. The basic result is a decrease in the steric repulsion above the coil-to-stretched chain transition and domination of attraction forces. Then, attraction forces promote the macromolecule concentration that is equivalent to the formation of the polymer phase and separation of a solvent. This result is in line with the understanding of the dilute solution in a static state as being ‘not dilute” in extension [171].

For ultrafine fibers, capillary forces must be taken into account. If the jet diameter becomes less than the contour chain length, the localization of the polymer phase inside the jet core makes it possible to cease a process of capillary break-up and to lead to the formation of the solvent annular droplets on a fiber [178,179].

#### Conclusions and Challenges 

The structure of polymeric liquids in a static state depends on its composition, space position of components and their molecular interactions. The rheological properties of a medium in this state are characterized by the frequency dependence of the complex modulus of viscoelasticity in the linear domain of small deformations (characterized by Newtonian viscosity or yield stress). Deformation in any geometry (in particular, simple shear or uniaxial extension) leads to changing this initial structure and the emergence of non-linear rheological behavior determined by the current structure of a matter. These changes depend on the evolution of macromolecular conformations, their orientation and interchain interactions that results in the occurrence of elasticity. All structure transformations happen in time and their scale can be rather long. The main rheological effects linked with deformation-induced structure evolution include thixotropy and yielding, phase transitions and instability at deformations.

Now the rheology of complex multicomponent fluids related to deformation-induced structure formation is considered a unified plasto-visco-elastic phenomenon. Various non-linear effects are understood as particular cases of general physical processes happening due to changes in a relaxation spectrum, relaxation or solid-to-liquid and liquid-to-solid phase transitions, bifurcations and instability in flow. Numerous experimental studies indicate to direct correlation between deformation-induced specific structure transformations and non-linear rheological properties of various polymeric liquids. General laws regulating these correlations cannot exist due to the great variety of real material, but such correlations undoubtedly exist in all cases, and there are mutual experimental and theoretical tasks of finding them in any specific situation.

Describing the modern state-of-the-art in the structure–rheology correlations, we should raise some general questions as to which are the open and remain challenges for further studies.

Although many experimental facts illustrate the general idea of the relationship between the structure of polymeric liquids and their rheological properties, in many cases, one can only state the existence of such relationships, but their quantitative description remains a challenge. This is especially true for the formulation of the conditions for shear-induced phase transitions. Another area of great interest for this issue is the role of heterogeneity and spatial distribution of components in the measurement of rheological parameters. 

Shear-induced anisotropy definitely affects the rheological properties of polymeric fluids. However, very little is known about the effect of shear on the anisotropy of properties. A separate aspect of this issue relates to solutions of rigid-chain polymers. This is an interesting problem of shear-induced phase transitions and their interrelationship with shear-induced birefringence. 

Does the statement “No steady state flows below the yield stress…” [57] have a universal meaning? Yes, this seems valid in many cases, but perhaps this strong statement depends on the definition of yield stress.

There are two different theoretical model for formulating the constitutive equations—tube and slip (time-dependent) entanglement model. It would be rather interesting to compare the predictions of both models and conclude which one of them is the most suitable and convenient for solving dynamic problems in the flow of complex rheological fluids.

The nature of deformation-induced phase separation apparently is not yet clear since it is necessary to understand the boundary between pure hydrodynamic (flow) processes and the stress-diffusion coupling mechanism. 

Many years ago, C. A. Truesdell opening the VIII International Congress on Rheology (1980), said: “Fortunately, today we hear less and less about “thixotropy”, more and more about constitutive equations”. This remains to be a rather debatable judgment. The current trend is to incorporate the kinetic equations (reflecting thixotropic effects) in the constitutive equations. It means that we imply the union of both fundamental concepts but not the exclusion of thixotropy, which continues to be a separate effect important for numerous applications. Does this issue continue to be debatable?

## Figures and Tables

**Figure 1 polymers-14-01262-f001:**
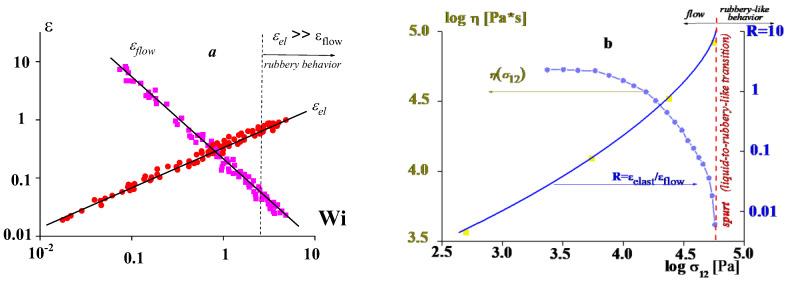
The relationship between fluidity and recoverable deformations in polyisoprene demonstrating the flow-to rubbery state transition, (**a**) and correlation with non-Newtonian flow (**b**). The ratio R is equivalent to the Weissenberg number [10].

**Figure 2 polymers-14-01262-f002:**
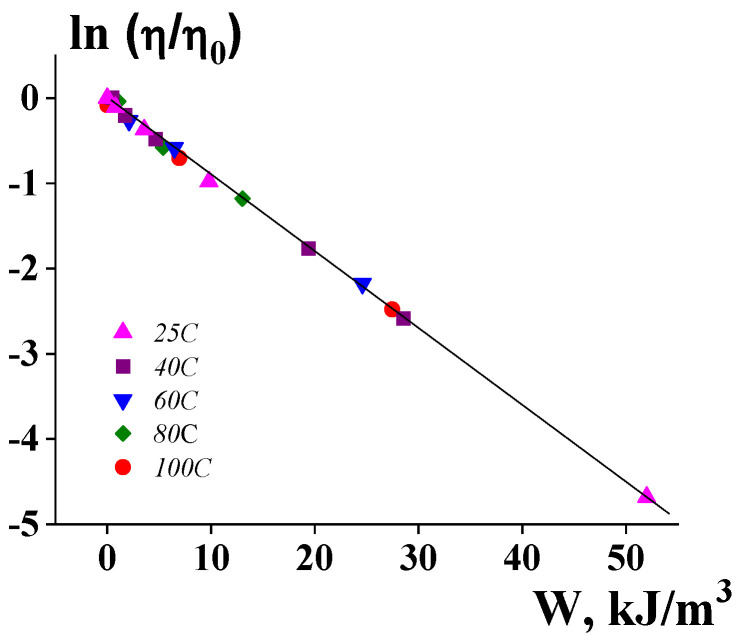
Dependence of the non-Newtonian effect of a polyisobutylene on the stored elastic energy at different temperatures [10].

**Figure 3 polymers-14-01262-f003:**
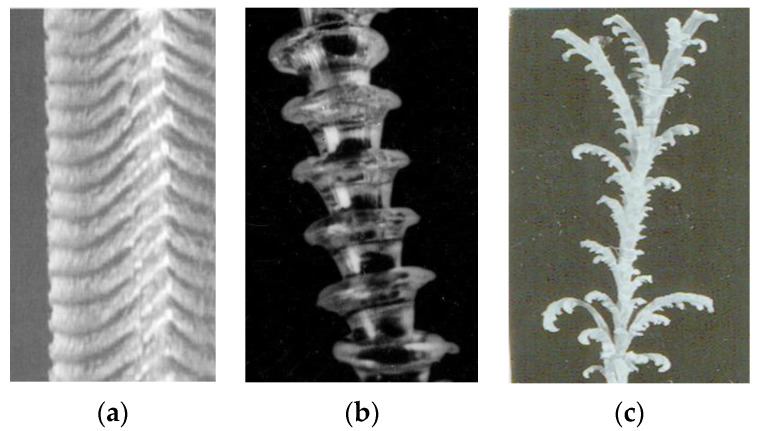
Different stages of elastic instability observed in increasing the Weissenberg number: (**a**) appearance of small-scale defects called “sharkskin”, (**b**) transition to strong periodic distortions, (**c**) developed elastic turbulence accompanied by melt fracture.

**Figure 4 polymers-14-01262-f004:**
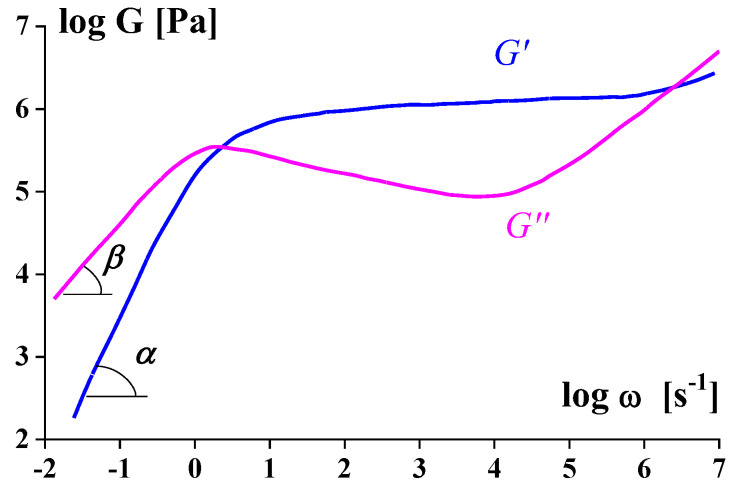
Typical frequency dependences of the components of the complex dynamic modulus for polymer melt (tan α = 2; tan β = 1).

**Figure 5 polymers-14-01262-f005:**
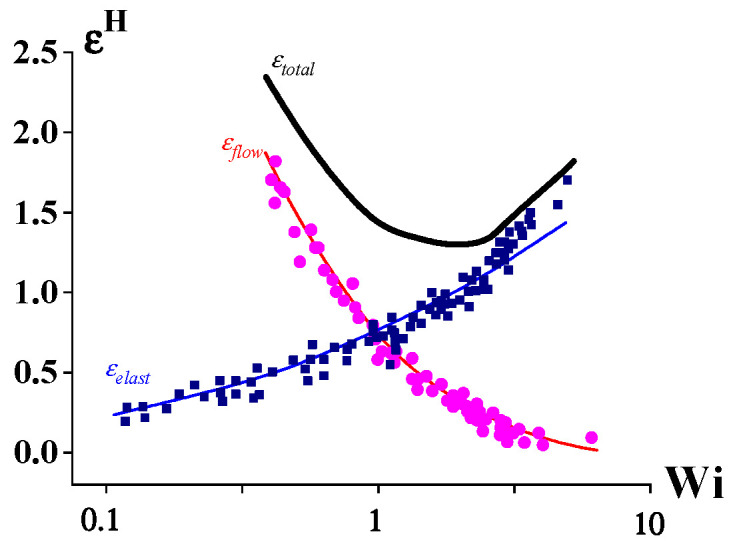
Dependences of the flow ($\varepsilon_{flow}$) and elastic ($\varepsilon_{elast}$) deformations on the Weissenberg number in uniaxial extension of uncured polyisoprene [10].

**Figure 6 polymers-14-01262-f006:**
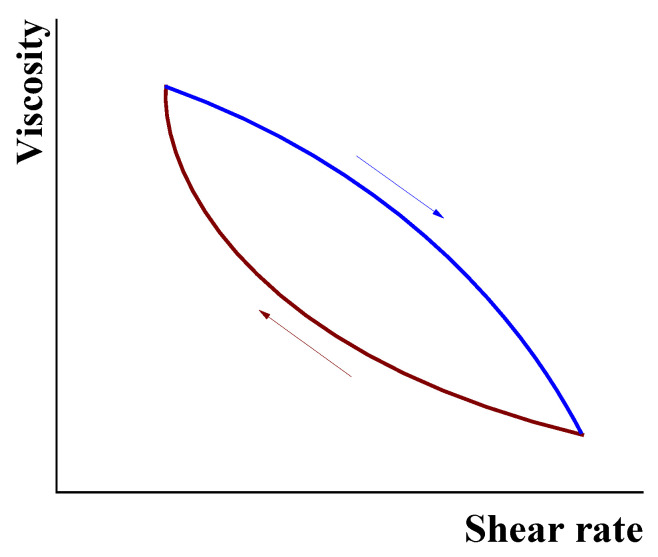
Typical thixotropic loop with complete recovery (return to the starting point).

**Figure 7 polymers-14-01262-f007:**
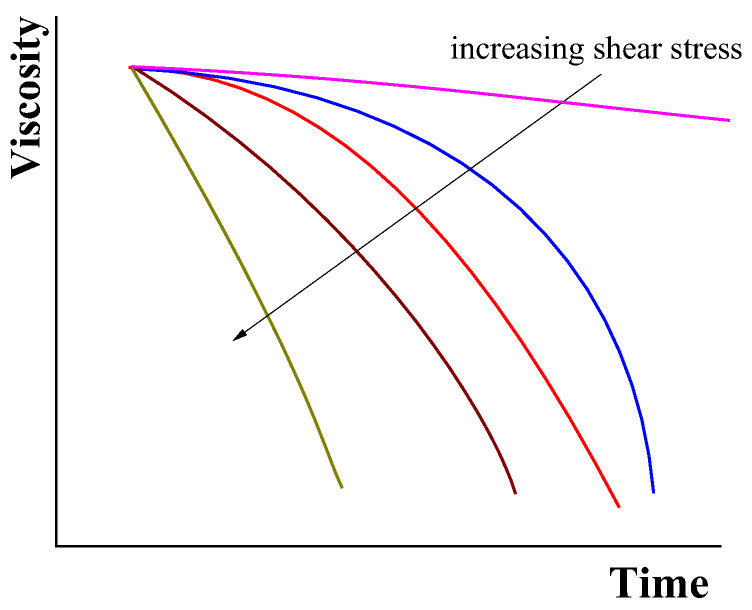
Thixotropic behavior demonstrated by the viscosity evolution at different shear stresses.

**Figure 8 polymers-14-01262-f008:**
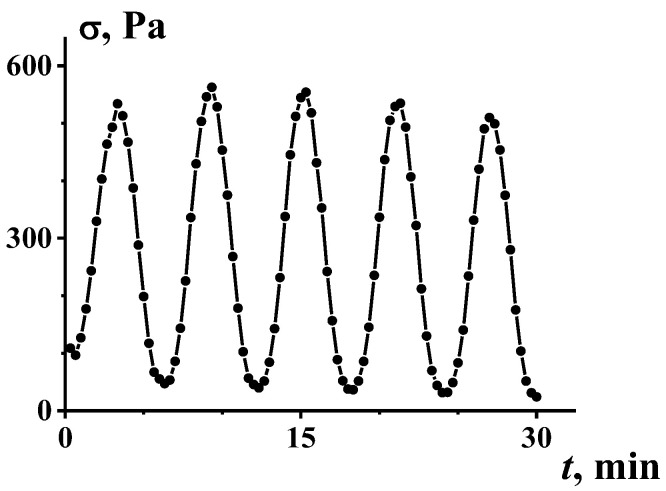
Stress self-oscillations in shearing concentrated suspension at $\dot{\gamma }=const$. Stress oscillations are equivalent to oscillations of the apparent viscosity.

**Figure 9 polymers-14-01262-f009:**
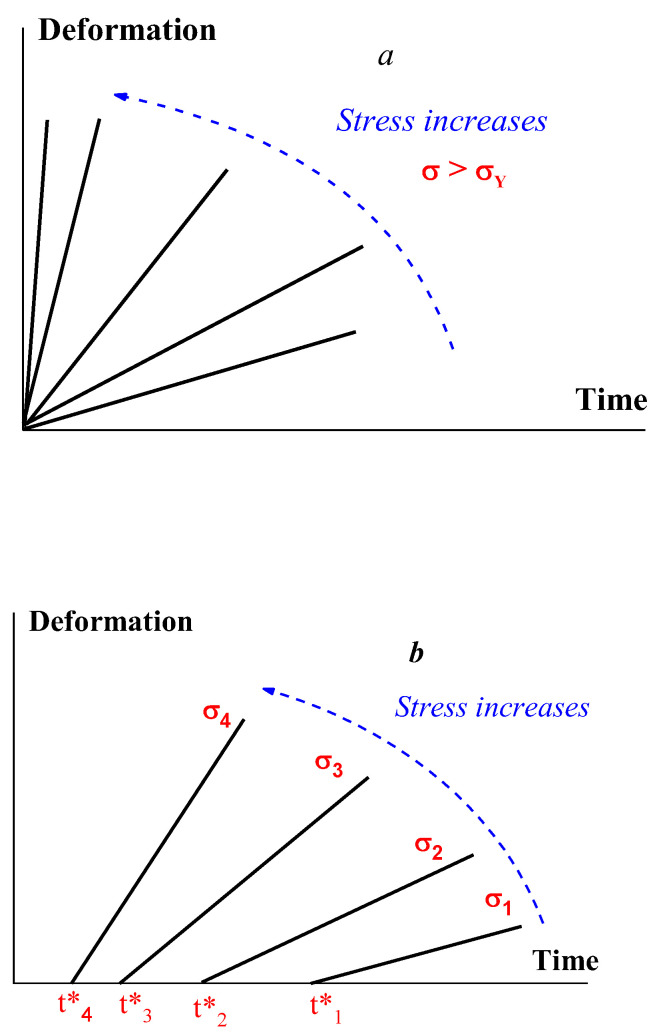
Yielding: leading to instantaneous solid-to-liquid transition at $\sigma >\sigma_{Y}$ (**a**) and durability as the dependence of the start of deformation on stress, t*(σ) (**b**).

**Figure 10 polymers-14-01262-f010:**
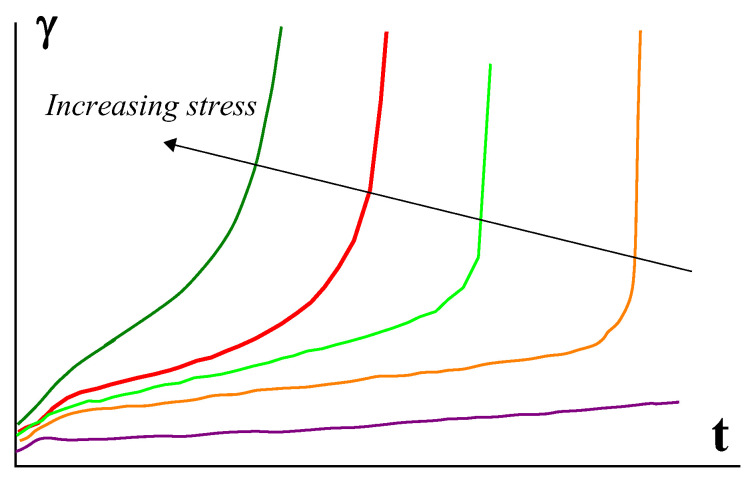
Time dependences of deformation for soft viscoplastic media.

**Figure 11 polymers-14-01262-f011:**
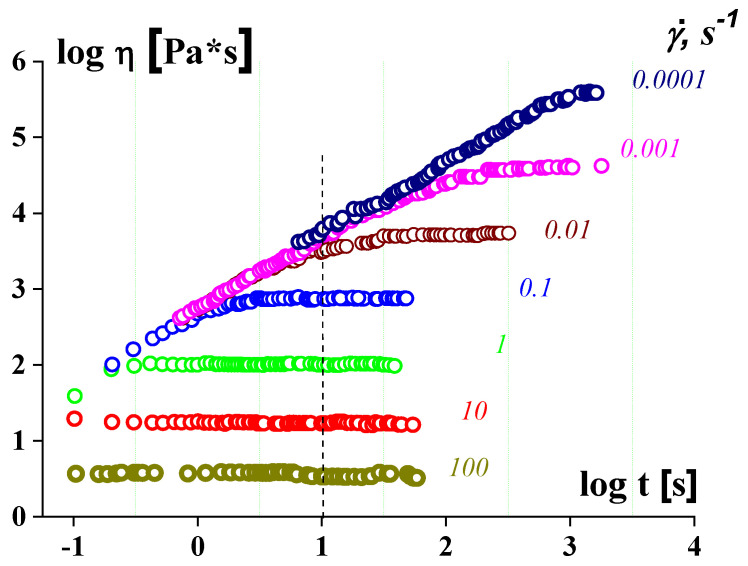
Time effects in measuring a flow curve in the scanning mode.

**Figure 12 polymers-14-01262-f012:**
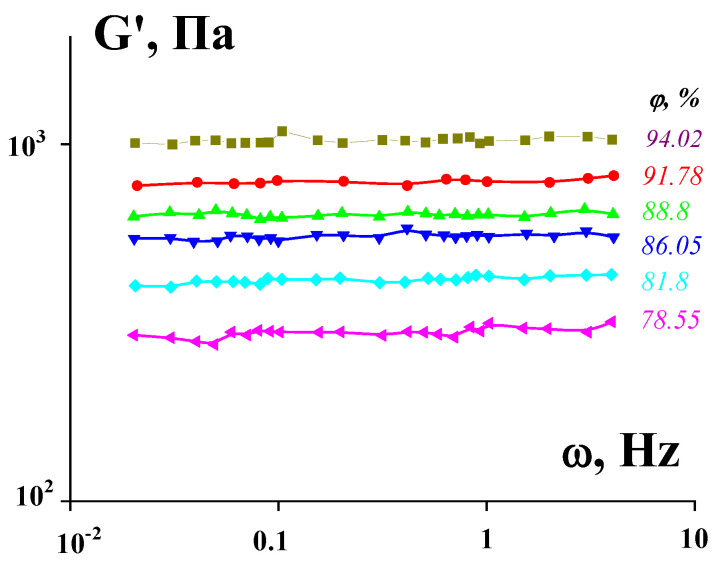
Frequency dependences of the elastic modulus of highly concentrated emulsions below the yield stress, (according to the experimental data presented in [63]). Concentrations are shown at the curves.

**Figure 13 polymers-14-01262-f013:**
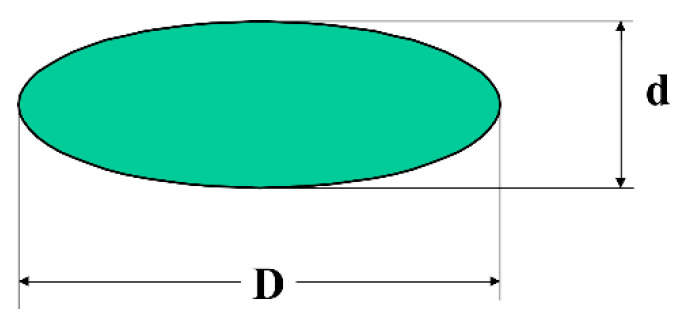
Anisometric solid particle.

**Figure 14 polymers-14-01262-f014:**
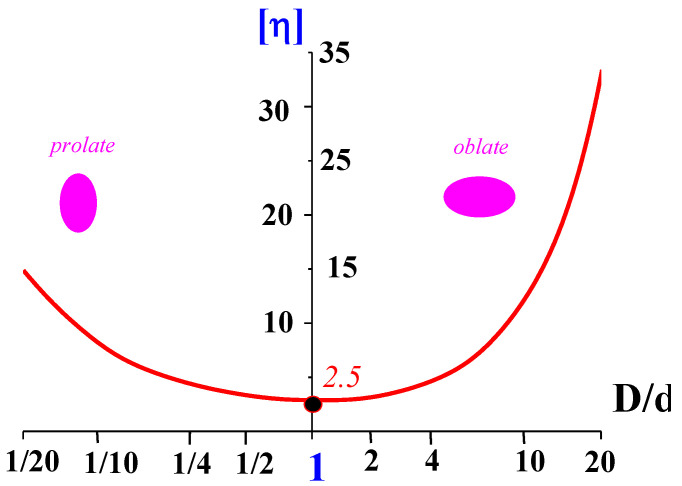
Influence of the anisotropy of solid particles on intrinsic viscosity (according to G.B. Jeffrey (1922) and W. Kuhn and H. Kuhn, (1945)).

**Figure 15 polymers-14-01262-f015:**
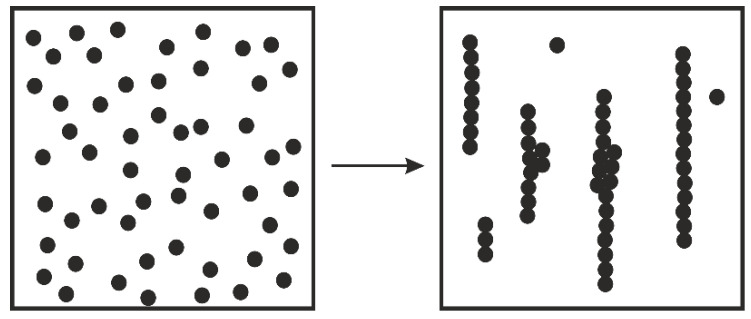
Self-assembling in transition from random (**left**) to regular chain-like (**right**) space distribution of disperse particles in a liquid phase.

**Figure 16 polymers-14-01262-f016:**
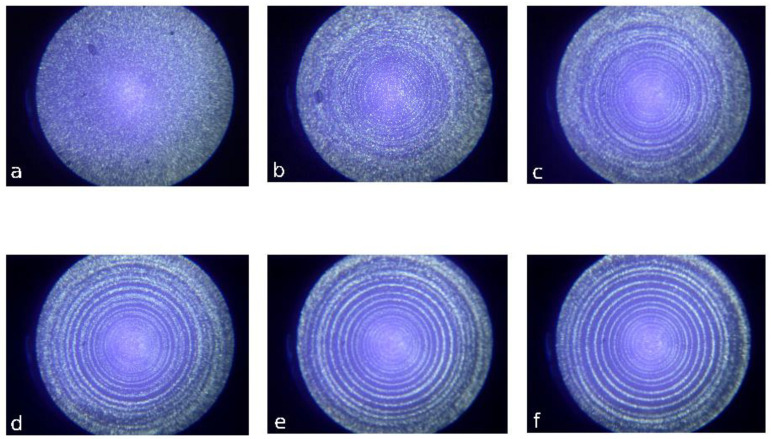
Consequent shots illustrating the kinetics of self-organization in rotational shearing of polyisobutylene solution filled with solid clay particles. Letters from (**a**–**f**) correspond to an increase in the time of shearing from 2 to 12 min.

**Figure 17 polymers-14-01262-f017:**
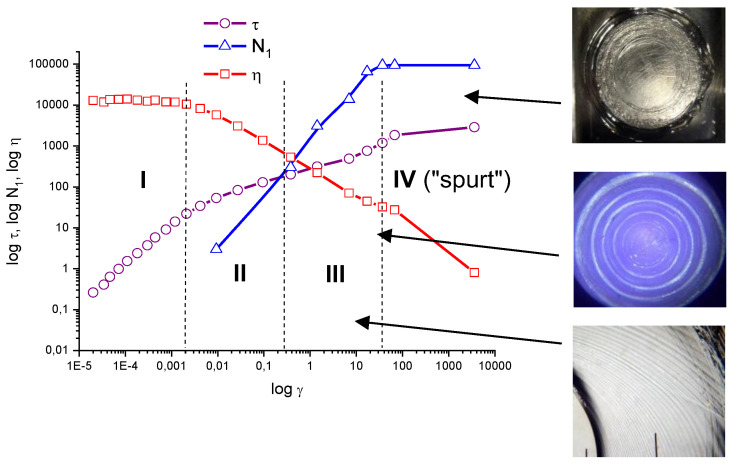
Rheological characteristics and morphology of stream for PIB solution filled with clay particles. Designation of zones: *I*—Newtonian flow, *II*—viscosity anomaly, *III*—area of dominant elasticity, *IV*—spurt behavior. Morphology of the sample is shown for zones *III* (**bottom** photo) and *IV* (**top** photo). 1E-5—1 × 10^−5^; 1E-4—1 × 10^−4^.

**Figure 18 polymers-14-01262-f018:**
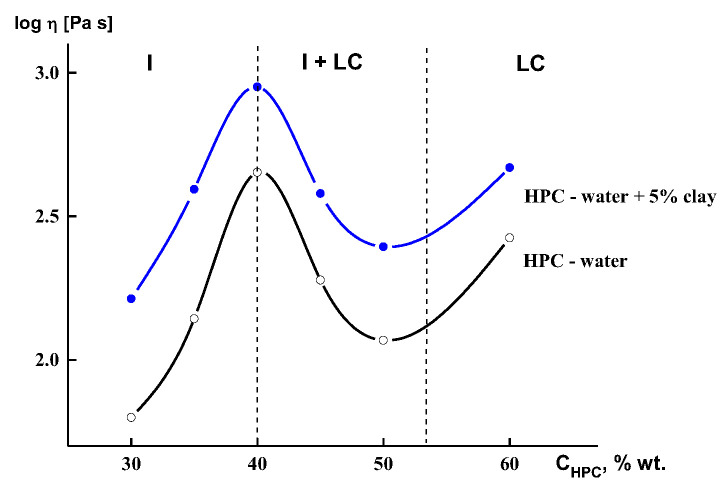
Concentration dependences of the viscosity of the neat and filled with 5% of clay (Na-montmorillonite) HPC solutions.

**Figure 19 polymers-14-01262-f019:**
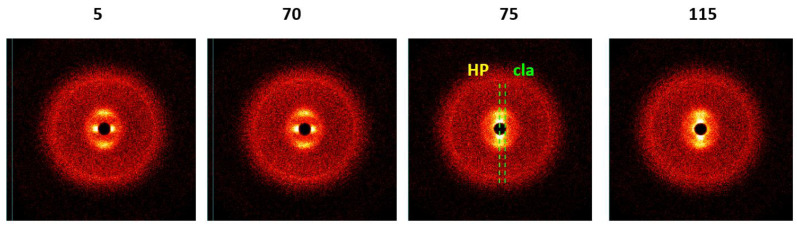
Evolution of X-ray diffractogram during prolong shearing of dispersion at shear rate of 471 s^−1^. Numerical values mean shearing time, min.

**Figure 20 polymers-14-01262-f020:**
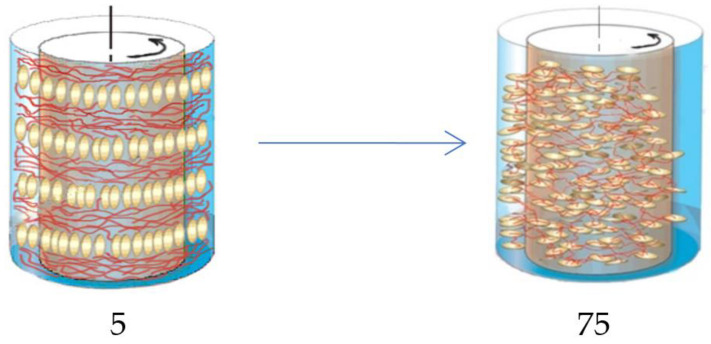
Evolution of clay particles ordering in shearing time, min, indicated by digits.

**Figure 21 polymers-14-01262-f021:**
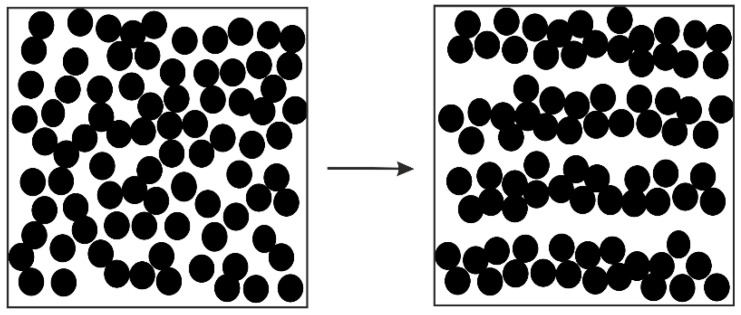
Self-assembling in high concentrated dispersions in the form of layered structure.

**Figure 22 polymers-14-01262-f022:**
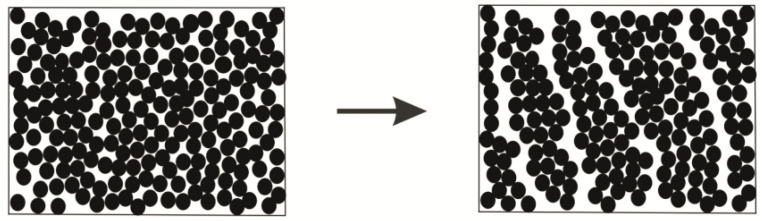
Formation of separate clusters of solid particles with their consequent displacement.

**Figure 23 polymers-14-01262-f023:**
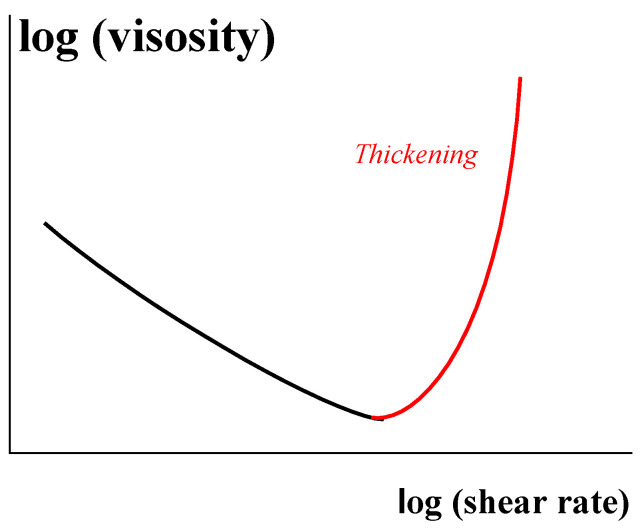
Transition from the viscosity decreasing at low shear rates to the thickening and jamming at high ones.

**Figure 24 polymers-14-01262-f024:**
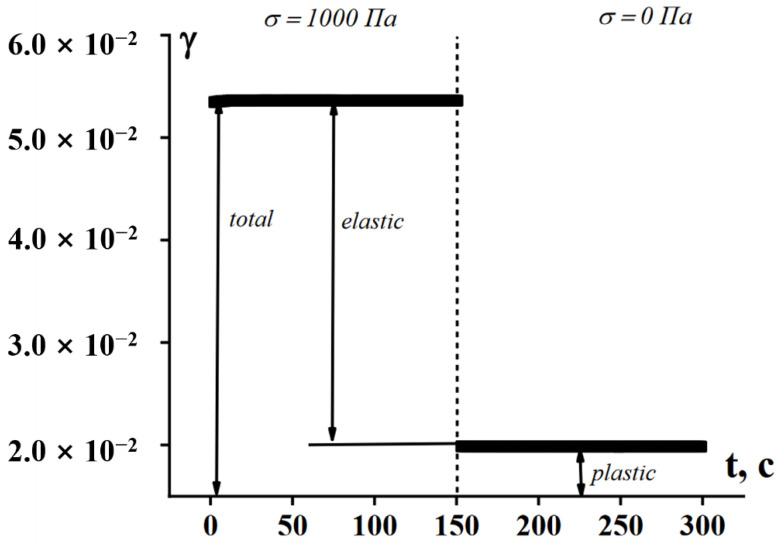
Elastoplastic behavior of a model feedstock based on 60% of Al_2_O_3_ in polymer matrix. Left part—shearing under stress of 1000 Pa; right part—rest (σ=0) after cessation of stress.

**Figure 25 polymers-14-01262-f025:**
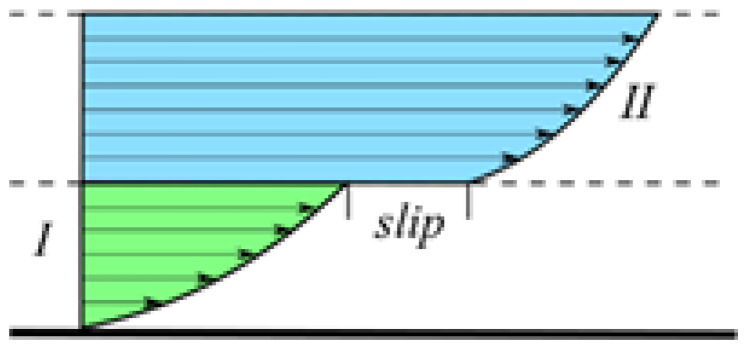
Velocity profiles in coextrusion of different polymer melts with a slip at the interface between polymers I and II.

**Figure 26 polymers-14-01262-f026:**
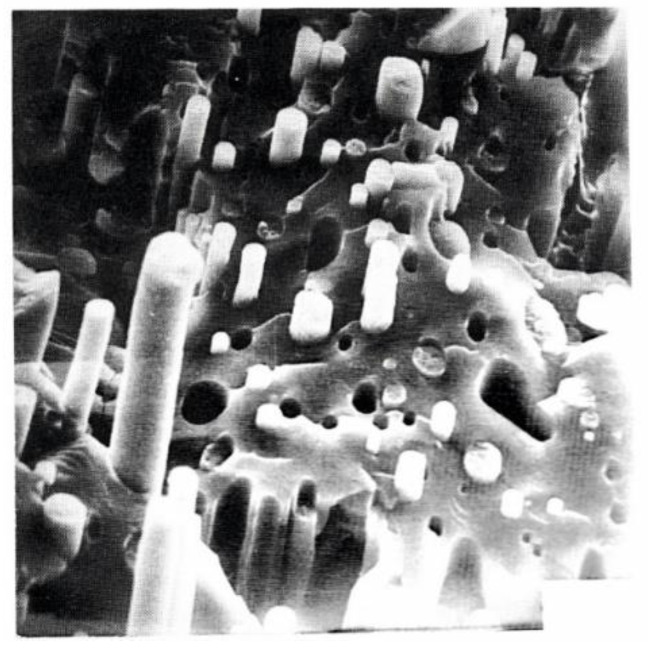
Cross section of the extrudate containing two incompatible polymers.

**Figure 27 polymers-14-01262-f027:**
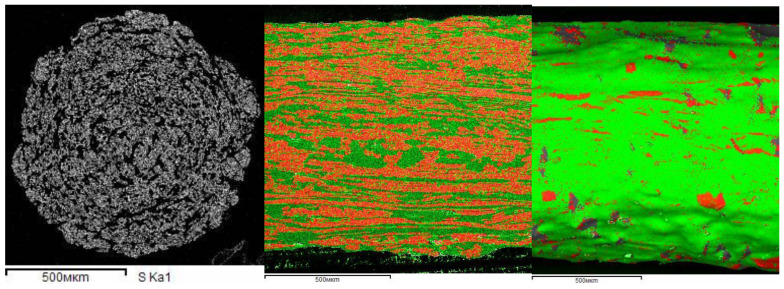
Cross section (**left**), longitudinal cutting (**middle**) and a surface (**right**) of an extrudate of the mixture PSF:LCP (50:50).

**Figure 28 polymers-14-01262-f028:**
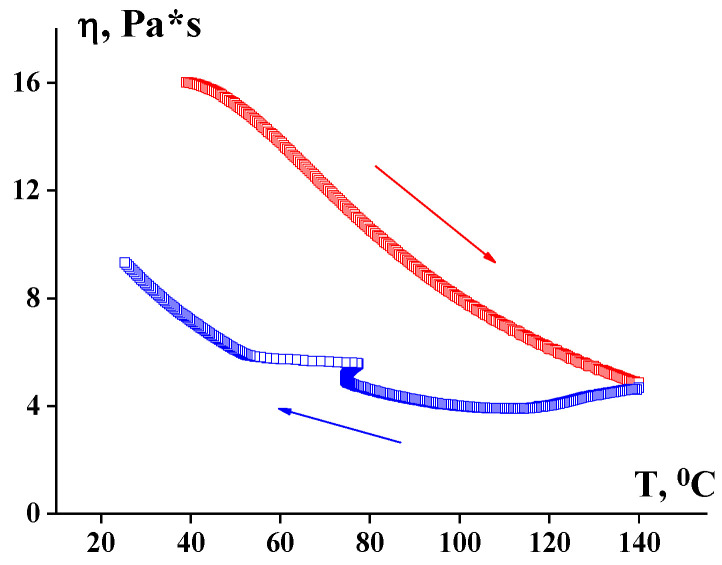
Thixotropy at temperature scanning for 4% solution of PABI in dimethylacetamide.

**Figure 29 polymers-14-01262-f029:**
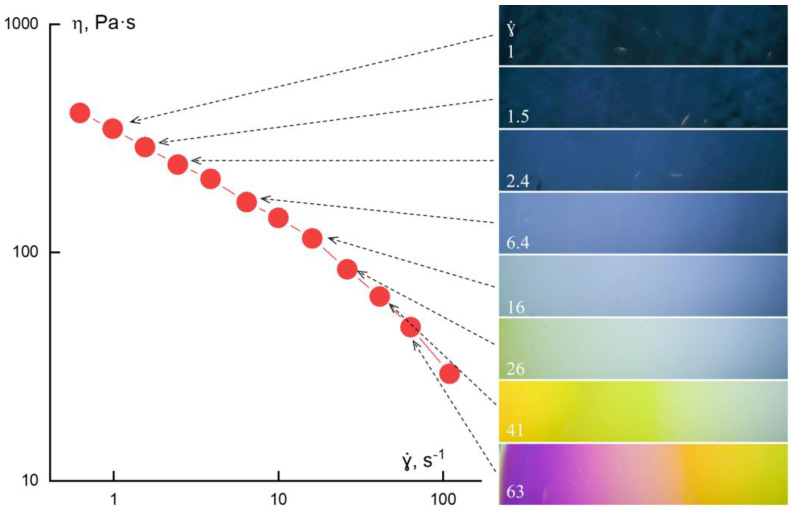
The flow curve of 5.7% solution of PABI in DMAc and the birefringence intensity corresponding to different shear rates (unpublished data presented by courtesy of Dr. Ivan Skvortsov).

**Figure 30 polymers-14-01262-f030:**
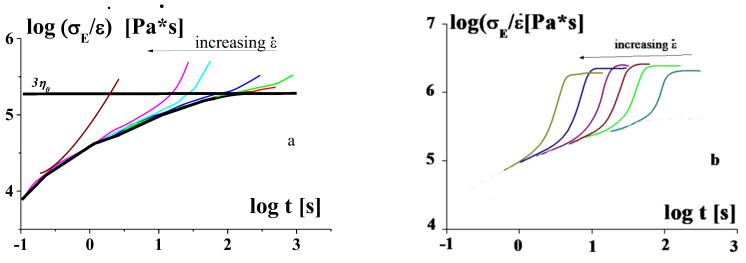
Typical examples of the rheological behavior of polymeric liquids (melts and solutions) in uniaxial extension (explanations are in the text).

**Figure 31 polymers-14-01262-f031:**
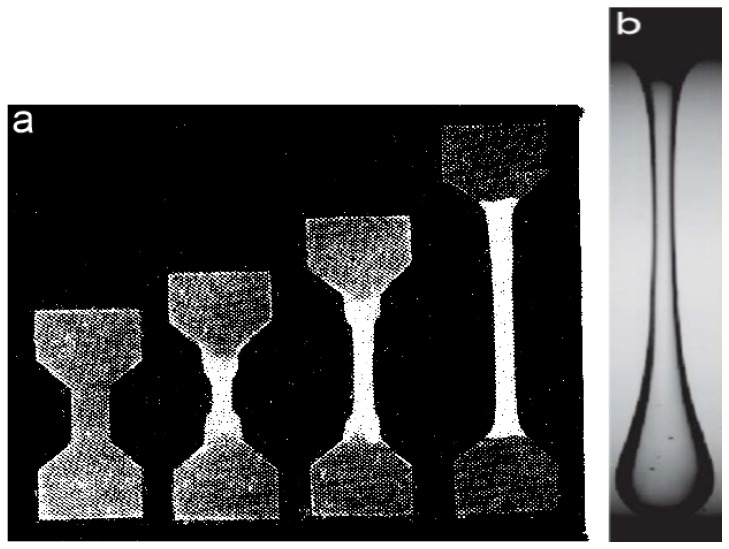
Heterogeneous stretching: necking (**a**) and gradual contraction (**b**).

**Figure 32 polymers-14-01262-f032:**
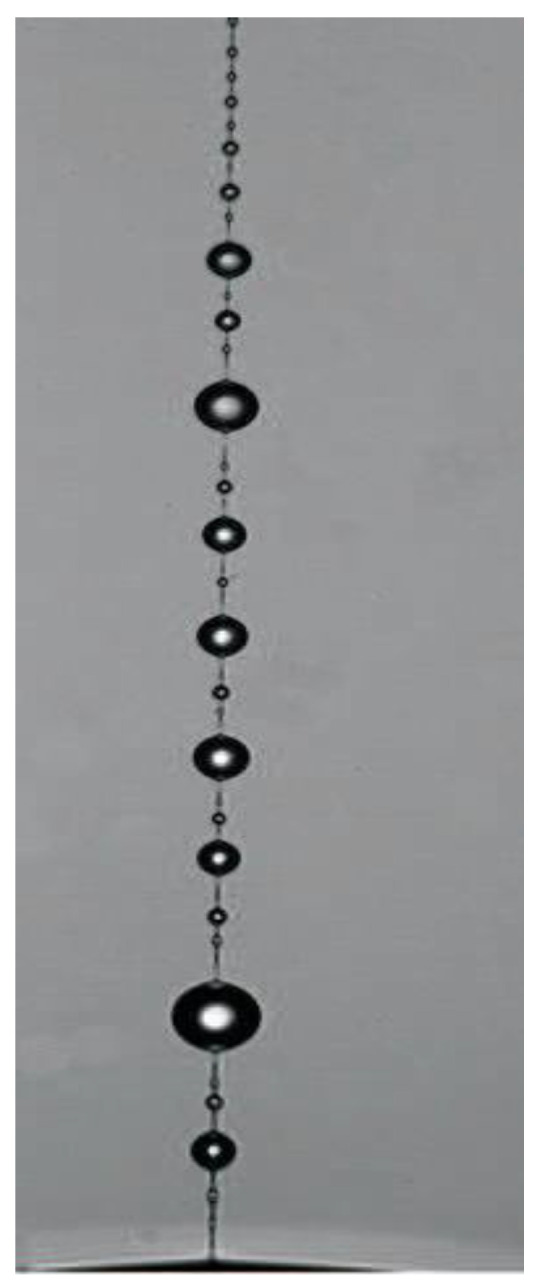
Beads-on-string structure occurring at stretching of polymer solutions.

**Figure 33 polymers-14-01262-f033:**
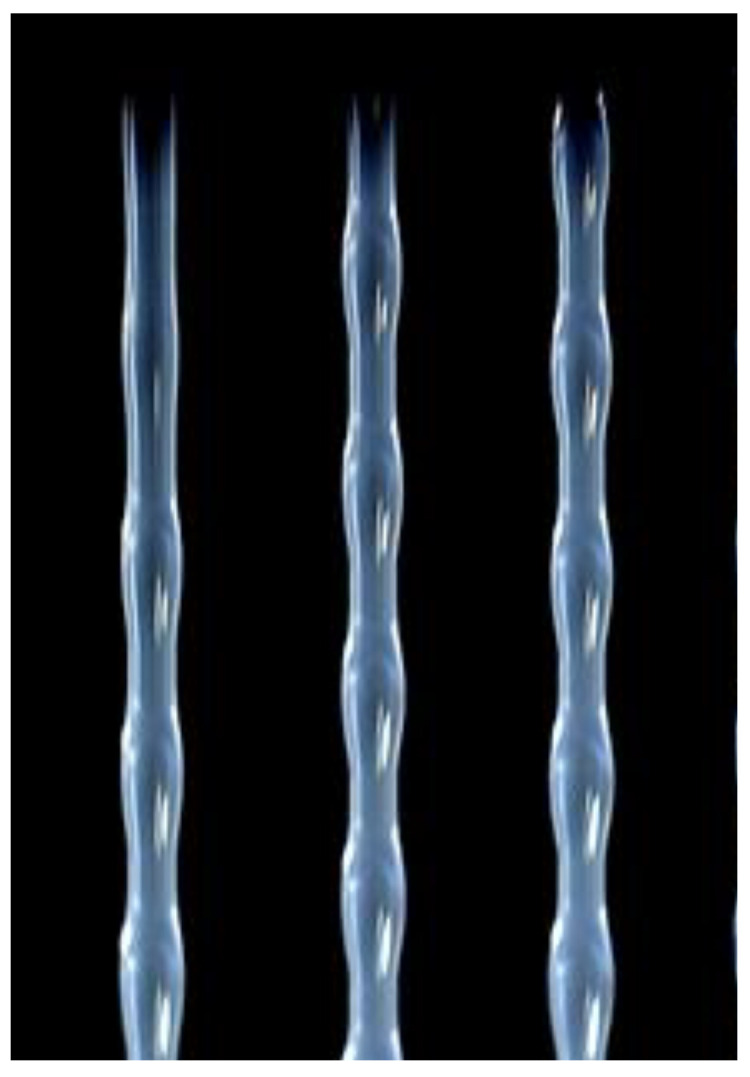
Formation of the bead-on-string structure—initial stage.

**Figure 34 polymers-14-01262-f034:**
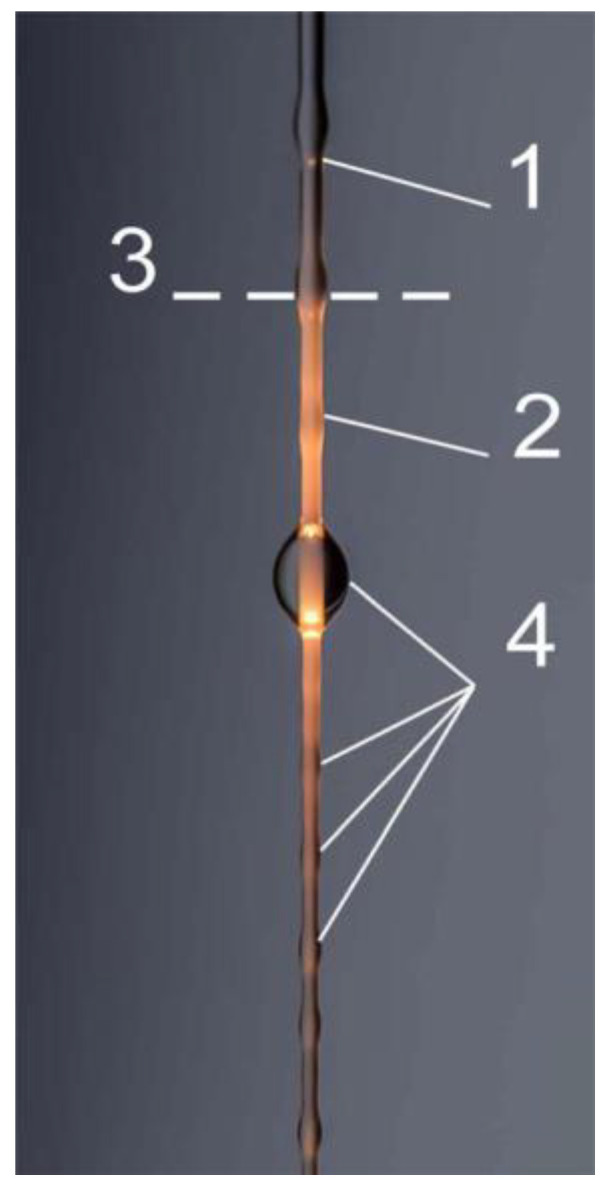
Phase separation in a stretched solution jet. 1—fluid part; 2—solidified part; 3—boundary of the phase separation detected due to appearance of light scattering on concentration fluctuations; 4—large and small solvent droplets spread over the filament surface.

## Data Availability

Not applicable.

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
