# Peer review of "The Role of Structure in Polymer Rheology: Review"

_polymers, 2022, doi:10.3390/polym14061262_

Round 1
Reviewer 1 Report
The authors reviewed recent studies on relations between the structure of the polymeric liquids on the rheology properties especially deep into the nonlinearity phenomena. This review covered nearly all aspect of modern research topics and provided a comprehensive overview and provide connection between these different topics, which is very inspiring. I highly recommend this review to be published on Polymers after the following minor format issues are being addressed.
- Typos: page 1, line 31, “prin.ciple” should be “principle”.
- Some words are labeled as deleted such as line 82: “at”, line 351: “a” & “transit to” and line 1111: “of”. Please make sure to deleted them in the final version.
- In Fig. 5, the curve of εtotal is disappeared after printed out, please change the color for clarity.
- In Fig. 17, the label of zone number should be I II III & IV, please revise it and make it consistent with the figure caption.
- Please reformat the Fig. 19 and Fig. 20 so the figures will not cover the figure caption.
Author Response
Thank you for detailed examination of the text and your comments. All of them meet the answers and corrections.
- Typos: page 1, line 31, “principle” should be “principle”
Corrected
.
- Some words are labeled as deleted such as line 82: “at”, line 351: “a” & “transit to” and line 1111: “of”. Please make sure to deleted them in the final version.
line 82 - deleted
line 351 – deleted
line 1111 –the text was corrected
- In Fig. 5, the curve of εtotalis disappeared after printed out, please change the color for clarity.
Now the color is black and we increased the width of this curve to make is more visible.
- In Fig. 17, the label of zone number should be I II III & IV, please revise it and make it consistent with the figure caption.
Corrected
- Please reformat the Fig. 19 and Fig. 20 so the figures will not cover the figure caption.
Done

Reviewer 2 Report
Please find attached my comments.

Author Response
The reviewer writes:
“If the authors can add a discussion paragraph or add comments to relevant sections about some of the open questions. Also, I found several typos, incomplete sentences, and a lack of lucidity in terms of reading the manuscript. In several parts of the review, there was a lack of continuity between paragraphs. Figures and their captions can also be improved. Some figures need a better resolution.
Thank you for very careful examination of the article. The most interesting is the proposal to add “about some of the open questions” This is really important and we have done it with pleasure. The following text has been added:
“Describing the modern state-of-art in the structure-rheology correlations, we should like raise some general questions which are the open and remain challenges for further studies.
Although many experimental facts illustrate the general idea of the relationship between structure of polymeric liquids and their rheological properties, in many cases, one can only state the existence of such relationships but their quantitative description remains a challenge. This is especially true for formulation the conditions for shear-induced phase transitions. Another area of great interest for this issue is the role of heterogeneity and spatial distribution of components in the measurement of rheological parameters.
Shear-induced anisotropy definitely affect the rheological properties of polymeric fluids. However, very little is known about the effect of shear on the anisotropy of propertiesю. A separate aspect of this issue relates to solutions of rigid-chain. This is an interesting problem of shear-induced phase transitions and their difference with shear-induced birefringence.
Does the statement “No steady state flows below the yield stress…” [57] have a universal meaning? Yes. this seems valid in many cases but perhaps, this strong statement depends on the definition of yield stress.
There are two different theoretical model for formulating the constitutive equations – tube and slip (time-dependent) entanglement model. It would be rather interesting to compare the predictions of both models and conclude which one of them is the most suitable and convenient for solving dynamic problems in the flow of complex rheological fluids.
The nature of deformation-induced phase separation apparently is not yet clear since it is necessary to understand the boundary between pure hydrodynamic (flow) processes and the stress-diffusion coupling mechanism.
Many years ago, C. A. Truesdell opening the VIII International Congress on Rheology (1980), said: “Fortunately today we hear less and less about “thixotropy”, more and more about constitutive equations”. This remains to be a rather debatable judgment. The current trend is to incorporate the kinetic equations (reflecting thixotropic effects) in the constitutive equations. It means that we imply union of both fundamental concepts but not the exclusion of thixotropy, which continues to be a separate effect important for numerous applications. Does this issue continue to be debatable?
All other comments were taken into account and we did all our best to make necessary corrections
Answers to additional comments
- Figure 29. The strain rate mentioned in the color bars showing birefringence intensity
does not match the arrows leading to the (red) data points in the viscosity v/s strain
rate plot. A few examples of the statements being ambiguous and confusing. (However, as mentioned above, I request the authors to proofread and improve upon major parts of the review to make the review more lucid and less ambiguous for the readers):
Corrected
- The sentence in Line 1010 “formulation of the criteria of this phenomenon in 154” –
which phenomenon are the authors talking about is not clear from the sentence?
Similarly, in the following line (Line 1012), it is not clear what is the purpose of this
sentence here? Are the authors talking about the theoretical models which help
determine the stability of jets or, are the authors mentioning that the experimental
rheological data helps determine the stability? The sentences do not bring out clearly
the meaning intended by the authors.
You are right. This paragraph was not clear. We reconstructed the text and changed the order of references.
- Line 1015: “Now, the role of viscoelasticity is included in theoretical and
experimental examination [155, 156]”. The reader is made to believe the authors are
claiming that recently viscoelasticity has played a critical role in the understanding of
the experimental data. However, the next line (Line 1016-1018) mentions “Just the
elasticity ….” – Do the authors mean that solely elasticity is sufficient and viscoelastic description is not required?
Of course, this is inaccurate. We should speak about viscoelasticity, not elasticity. Corrections have been done.

Round 2
Reviewer 2 Report
I thank the author for the revised version. It was a pleasure reading the revised manuscript. I recommend the manuscript for publication. Also please find below a few comments.
Comments:
- Line 290: Please refer to the correct equation number instead of (2).
- Line 310: Citation [43] does not seem to be the appropriate reference (as it is not a review). Also, the doi link of [43] is incorrect in the references section. Please kindly review and check the references and citation numbering.
- Line 310: Please rephrase the sentence: “The authors of these comprehensive reviews….”. ” Only a single review i.e. [43] is mentioned in the previous sentence.
- Line 317-318: “......, that is why there is no sense to repeat them here.” I suggest the authors remove this part of the sentence.
- Line 352-354: Please rephrase the sentences. Furthermore, I request that the authors use a different word instead of the word "gimmicks" in their description of the cited peer-reviewed article since the word "gimmick" is generally associated with a negative connotation that implies concealment.
- Line 626: Caption in Fig. 19 ‘Figures mean shearing time, min.’ I believe the authors are referring to the ‘numerical values’ in the figures. I request the authors to elaborate on the sentence.
Please rephrase/ check the grammar of the following sentences/paragraphs:
- Line 169-171: “This is a case ….take place”
- Line 182-186: “The fail the Cox-Merz …. Stronger non-linearity”
- Line 192: “as one of the historical milestones in rheology”. Should ‘as’ be ‘is’?
- Line 226: ‘but do not have a physically based ground’- I suggest adding a hyphen ‘’physically-based” or replacing “physically based ground” with “ physical basis”.
- Line 279: ‘..... for the dominance different types ……’
- Line 306: ‘Today’s understanding the different manifestations of thixotropy…….’
- Line 396-398: ‘However, in the scanning mode…..Newtonian plateau’.
- Line 412: “at all”
- Line 439-441: “The convincing experiment….(the Weissenberg effect)”
- Line 444-445: “In this case, ….., if it was”
- Line 666-668: “The effect of layering …….to separation of components.”
- Line 687 - 690: “There is a lot of …..(Fig. 23).”
- Line 703: “Meanwhile…plastic deformations.”
- Line 745-747: “Indeed, …..crystallization.”
- Line 796 -797: “In addition, ……capillary.”
- Line 907- 909: “It should be noted…Trouton law:”
- Line 1007: “There is ….both,”
- Line 1011-1014: “Perhaps, ….curvilinear of channels”
Please correct the spellings in the following lines:
- Line 37: ‘dimensional’
- Line 38: ‘accurately’
- Line 85: ‘Non-Newtonian’
- Line 160: ‘Cox-Merz’
- Line 275: ‘Maxwell’
- Line 482: ‘created’
- Line 659: ‘Similar’
- Line 676: ‘destroys’
- Line 701: ‘dense’
- Line 784: ‘polymer’
- Line 1083: ‘Challenges’
- Line 1088: ‘yield stress’
- Line 62: Remove the space between ‘th’ and ‘e’.
- Line 68: Delete ‘like’ from ‘......look like equivalent….’
- Line 115: Please insert a period after melt.
- Line 158: Delete of from the sentence “ Nevertheless, even of non-linearity …… ”
- Line 162: ‘Remove ‘.’ from two dots.
- Line 164: ‘for at ….’ please correct the sentence
- Line 190: Please insert a space between ‘Wo.Ostwald’
- Line 228: Please insert a space between ‘startand’
- Line 256: Delete ‘to’ from “to its equilibrium level….”
- Line 271: Please use proper punctuation after ‘plastic behavior’
- Line 383: The word ‘Soft’ is incomplete.
- Line 815: Please replace ‘meet’ with ‘met’
- Lie 1055: Please delete ‘do not’ as it has been repeated twice.
Author Response
Dear reviewer,
We appreciate your detail consideration of this paper. Your competence and deep analysis of the text including not only scientific content but also spelling evoke a feeling of deep gratitude. All your comments are accepted, and the text is corrected.
Thanks a lot!